# The expanding roles of homologous recombination proteins in genome stability

Lorenzo Sassi [1,2], Andrea Martinez Marroquin [1,2], Salli Waked [1,2], Alessandra Ardizzoia [1,2] & Vincenzo Costanzo [1,2 ✉]

## Abstract

**Homologous recombination (HR) is traditionally portrayed as a DNA double-strand break repair pathway. However, emerging evidence positions RAD51, its partners BRCA1, BRCA2, and other HR factors at the core of a broader genome-maintenance network that operates by a "prevent and protect" strategy extending beyond repair. Here, we review how RAD51 can shield DNA from nucleolytic processing mediated by MRE11 and related nucleases, promote fork reversal, suppress replicative DNA gaps accumulation, and bind abasic sites, averting their conversion into cytotoxic intermediates. These extended functions counteract endogenous replication stress as shown in BRCA1- or BRCA2-deficient contexts, where failure to prevent gaps, protect forks, and safeguard abasic DNA accelerates genomic instability. The functional impairment of HR proteins, which interface with base-excision repair and translesion synthesis, rewires these pathways, driving distinctive base-substitution mutational signatures of HR-defective tumors. Abasic sites, especially from methyl-cytosine metabolism, put replication forks at risk of breaking, amplifying the need for RAD51-mediated defense. Such redefinition of homologous recombination protein function as part of an anticipatory surveillance and protective system, rather than a repair-only module, bears important implications for understanding tumorigenesis, therapy resistance, and aging.**

**Keywords** RAD51; BRCA2; MRE11; Abasic Sites; Replication Fork Protection
**Subject Category** DNA Replication, Recombination & Repair

## Introduction

Homologous recombination (HR) is an accurate DNA repair pathway important for preserving genomic stability. Its primary role is to repair DNA double-strand breaks (DSBs), one of the most deleterious types of DNA damage, which mainly arise from replication stress, oxidative lesions, or exposure to ionizing radiation. HR is of uniquely high fidelity, using a homologous sequence, typically from the sister chromatid, as a template to restore the original DNA sequence with precision. This ensures accurate repair and faithful transmission of genetic information, underscoring the vital genome-safeguarding function of HR.

HR-mediated DSB repair is a highly orchestrated process that begins with the recognition and resection of DNA ends, generating single-stranded DNA (ssDNA) overhangs. This resection step commits the repair process to HR, diverting it from error-prone alternatives like non-homologous end joining (NHEJ). Initially, ssDNA is bound by replication protein A (RPA), which prevents secondary structure formation. RPA is then replaced by RAD51, a central recombinase that polymerizes on ssDNA to form a nucleoprotein filament. This filament facilitates homology search and strand invasion, leading to the formation of a displacement loop (D-loop) in the homologous template, thereby enabling DNA synthesis and restoring genomic integrity (Jasin and Rothstein, 2013; Paques and Haber, 1999; Szostak et al, 1983; West, 2003).

Defects in HR components such as BRCA1, BRCA2, RAD51, and RAD51 paralogs XRCC2, XRCC3, RAD51B, RAD51C, and RAD51D impair this high-fidelity repair process. HR-defective cells rely on alternative mechanisms like NHEJ and microhomology-mediated end joining (MMEJ), which introduce insertions, deletions, and chromosomal rearrangements (Schrempf et al, 2021; Suwaki et al, 2011). Such genomic aberrations contribute to a mutator phenotype and drive tumorigenesis. Experimental models, including mice carrying mutations in HR genes BRCA1 and BRCA2, recapitulate some features of human cancers, such as spontaneous tumor formation and chromosomal instability (Connor et al, 1997; Jonkers et al, 2001; Xu et al, 1999), establishing a direct causal link between HR deficiency and cancer development.

Intriguingly, emerging evidence reveals that HR proteins act beyond canonical DSB repair. This includes roles in replication fork stability and remodeling, gap suppression, and protection of vulnerable DNA intermediates. Although they are not yet completely understood, these extended functions help to explain a number of HR-deficient cell features that cannot be exclusively linked to defective DSB repair. A major aspect of HR-defective cells is the spontaneous occurrence of replication stress, a defining trait of many cancers (Halazonetis et al, 2008). During DNA replication, stalled replication forks are prone to collapse in the absence of functional HR proteins (Lomonosov et al, 2003; Yu et al, 2000). This leads to the accumulation of further DNA damage and the emergence of mutations typically found in HR-deficient tumors.

[1]IFOM-ETS, The AIRC Institute of Molecular Oncology, Milan, Italy. [2]Department of Oncology and Hematology-Oncology, University of Milan, Milan, Italy.
✉E-mail: Vincenzo.Costanzo@ifom.eu

Advances in methodologies, including DNA fiber analysis and DNA electron microscopy (EM), have enabled the identification of HR-related replication-associated events that occur even before DSB repair (Hashimoto et al, 2010; Schlacher et al, 2011). Several studies have revealed the formation of ssDNA gaps at unchallenged replication forks and the degradation of nascent DNA at stalled and reversed forks, which are processes in part mediated by nucleases such as MRE11, EXO1, and DNA2, and regulated by BRCA1 and BRCA2 (Cassani et al, 2018; Cong et al, 2021b; Cotta-Ramusino et al, 2005; Gonzalez-Prieto et al, 2013; Halder et al, 2022; Hashimoto et al, 2010; Kolinjivadi et al, 2017a; Kolinjivadi et al, 2017b; Menin et al, 2018; Mijic et al, 2017; Paes Dias et al, 2021; Schlacher et al, 2011; Sogo et al, 2002; Vugic et al, 2023; Zellweger et al, 2015). Additional studies have linked these fork-proximal activities to suppression of lagging-strand ssDNA gaps, reinforcing the concept that the HR pathways govern replication-borne lesion avoidance beyond DSB repair (Cong et al, 2021a; Vugic et al, 2023).

More recently, broken DNA replication structures with asymmetric forks have been observed by electron microscopy of DNA (DNA-EM) upon spontaneous breakage occurring in the absence of functional HR factors and polymerase theta (POLθ) or active BER enzymes such as APEX1 (Hanthi et al, 2024; Mann et al, 2022). These structures resemble forks with DSBs induced by selective cleavage of leading or lagging strands with CRISPR/Cas9 (Elango et al, 2025; Kimble et al, 2025; Pavani et al, 2024; Scully et al, 2024).

Overall increased processing of nascent and template DNA at replication forks indicate that HR proteins not only facilitate DSB repair, but also play a fundamental role in preventing the formation of toxic lesions and protecting unstable replication intermediates. Defects in these pathways have profound implications for cancer development. These recently identified functions of HR proteins contribute to explain characteristic base-substitution mutational signatures and enhanced sensitivity to replication stress-inducing therapies of HR-deficient tumors (Degasperi et al, 2022; Polak et al, 2017). They also help to delineate the high efficacy of PARP inhibitors, which exploit the synthetic lethality between HR dysfunction and impaired single-strand break (SSB) repair (Caldecott, 2024; Cong et al, 2021b; Huang and Kraus, 2022).

Altogether, the widening set of non-canonical HR activities reveals a broader genome-maintenance program centered on lesion prevention and fork integrity maintenance. These functions operate on intact active or stalled replication intermediates, before DNA breakage and fork collapse occur. They are mechanistically distinct from canonical DSB repair, which can follow fork breakage and requires end resection, RAD51 filament formation on ssDNA, strand invasion, homology search, and repair synthesis. Here, HR factors instead limit the accumulation of replication-coupled lesions, suppress ssDNA gaps, and protect vulnerable fork structures from nucleolytic attack, thereby preserving restart competence. Defining these roles clarifies the cellular consequences of HR deficiency even in unchallenged conditions, explains why HR proteins are essential, and provides a framework for therapies that selectively exploit the vulnerabilities of HR-deficient cancers. Here, we highlight these underappreciated protective and preventive functions of HR proteins, and how their loss contributes not only to cancer and therapy resistance but also to epigenetically driven replication stress and aging.

# HR proteins in the prevention of DNA lesion accumulation

## ssDNA gaps and abasic sites

ssDNA gaps have been observed during DNA replication in cells deficient in RAD51, BRCA1, BRCA2, and other HR factors (Cong and Cantor, 2022; Feng and Jasin, 2017; Hashimoto et al, 2010; Kolinjivadi et al, 2017b; Taglialatela et al, 2021). ssDNA gaps on template DNA can lead to replication fork collapse, chromosomal aberrations, and cell death, thereby contributing to cancer progression and therapeutic resistance. Consistent with a direct role for HR proteins in DNA replication, defects in RAD51 and BRCA2 affect the maturation of Okazaki fragments during lagging-strand synthesis, resulting in persistent ssDNA gaps accumulating on the lagging strand (Mann et al, 2022; Thakar et al, 2022). In addition, BRCA-deficient cells often fail to adequately restrain DNA replication fork progression in response to stress, further promoting the accumulation of ssDNA gaps (Cong et al, 2021a). This unrestrained replication exacerbates genomic instability and contributes to the hypersensitivity of BRCA-deficient cancers to genotoxic therapies.

Experiments based on DNA-EM identified ssDNA gaps on DNA templates damaged by UV irradiation in yeast (Lopes et al, 2006). Such gaps occur because replication forks that encounter UV-induced lesions can become uncoupled, with helicase-driven unwinding continuing even when leading strand synthesis stops. Long stretches of ssDNA build up on the leading strand side, putting the replication fork at risk and potentially compromising the integrity of the genome.

The first direct observation of ssDNA gaps during DNA replication in the absence of RAD51 came from *Xenopus laevis* egg extracts, where RAD51 was found to be essential for preventing the accumulation of ssDNA gaps at replication forks and behind them, even in the absence of exogenous DNA-damaging sources (Hashimoto et al, 2010). Replication forks in RAD51-depleted extracts show widespread uncoupling of leading- and lagging-strand synthesis, which leaves ssDNA gaps at the fork junctions (Fig. 1A). This observation implies that RAD51 engages with forks, possibly on the lagging strand, which harbors longer tracts of ssDNA, and promotes the re-annealing of the unwound templates. The prevalence of RAD51 on the lagging strand is compatible with experiments showing that an excess of lagging-strand polymerase α (POLα) displaces RAD51 bound to replicating chromatin (Mann et al, 2022). The absence of RAD51 also causes gaps behind forks located distantly from the fork junction (Fig. 1B). These gaps are further increased by DNA-alkylating agents and enlarged by MRE11-dependent degradation of newly synthesized DNA strands, whereas ssDNA gaps at fork junctions are largely independent of MRE11 activity. These findings directly suggested that gaps generated behind forks require stable RAD51 binding in order to prevent their resection (Fig. 1B), a notion confirmed by more recent observations made in BRCA1-defective cells (Seppa et al, 2025). Gaps behind forks can be formed by the polymerase PrimPol, which is able to skip template lesions and continue DNA synthesis after having synthesized its own RNA primer (Fig. 1B) (Bianchi et al, 2013; Garcia-Gomez et al, 2013; Mouron et al, 2013). The original work in *Xenopus laevis* highlighted two previously

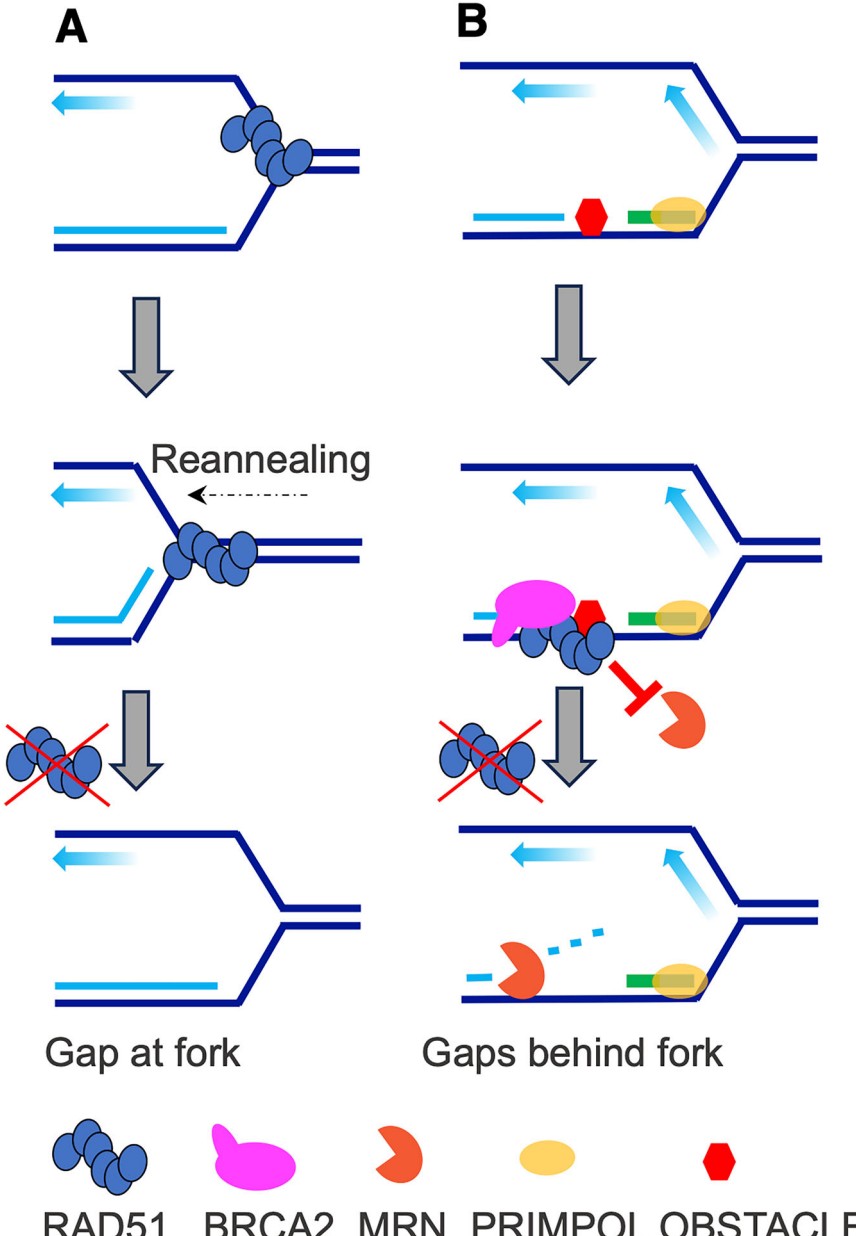

**Figure 1. HR-independent functions of RAD51 at replication forks.**

(A) RAD51 suppresses ssDNA-gap formation at forks by promoting re-annealing of transiently uncoupled parental strands. (B) BRCA2-stabilized RAD51 filaments protect gaps generated behind forks at sites of stalling and re-priming, preventing their nucleolytic resection and limiting gap accumulation.

unknown functions for RAD51, namely ensuring continuous DNA synthesis by prevention of spontaneous gaps, and safeguarding nascent DNA from nucleolytic degradation. Similar findings were subsequently reported for BRCA2-depleted egg extracts (Kolinji-vadi et al, 2017b) and human BRCA1-defective cells (Taglialatela et al, 2021). Additional links between RAD51 and DNA replication control come from the implication of RAD51 in restricting DNA over-replication following forced activation of DNA replication origins (Munoz et al, 2024).

Overall, these observations highlight the role HR proteins in preventing the occurrence of abnormal forks. However, several

questions remain unanswered. Among these there are the following: (1) which are the lesions spontaneously occurring during DNA replication that cause frequent polymerase stalling? (2) Do these lesions also occur in HR-proficient cells and are subsequently repaired by HR? (3) Or, do these lesions occur at higher rates in HR defective cells? The classical vision is that during DNA replication, replication forks stall when they encounter DNA lesions, such as bulky adducts, oxidative damage, or cross-links. This stalling results in the formation of ssDNA gaps as the leading strand continues synthesis past the lesion through re-priming operated by PrimPol, while the lagging strand continues

unimpeded (Quinet et al, 2020). HR could repair these gaps by utilizing the intact sister chromatid as a template to accurately restore the missing sequence in a process called template switch (TS). In particular, in the presence of persistent DNA lesions induced by bulky agents, PrimPol-generated gaps are subjected to BRCA1- and BRCA2-dependent repair (Nagaraju and Scully, 2007; Piberger et al, 2020). In HR-deficient cells, however, the inability to use TS and other HR-dependent DNA strand exchange-based pathways, unrepaired ssDNA gaps accumulate. Such gaps are highly vulnerable to degradation or further damage if not resolved by alternative, error-prone pathways, which at the same time introduce mutations and contribute to tumorigenesis.

Still, besides this straightforward explanation, additional mechanisms might contribute to the extensive accumulation of ssDNA gaps in HR-defective cells. As ssDNA gaps occur at high rates, it is unlikely that they are only due to defective TS-mediated repair, given the relatively low efficiency of HR-dependent reactions: DNA-EM does not reveal spontaneous TS intermediates in HR-proficient cells, unless these cells are treated with alkylating agents and genomic DNA is selectively enriched by 2D-gel purification (Giannattasio et al, 2014). The intrinsically slow step of DNA pairing in TS reactions is incompatible with the rapid kinetics of DNA replication. It is therefore more likely that HR proteins prevent spontaneous lesions that lead to polymerase stalling and ssDNA gaps, in addition to repairing them.

Among the most frequent spontaneous polymerase-stalling lesions that could give rise to ssDNA gaps are abasic, aka apurinic/apyrimidinic (AP), sites (Lindahl, 1993; Lindahl, 2004). AP sites are among the most common forms of DNA damage, resulting from either spontaneous hydrolysis of the glycosidic bond linking the base to the sugar-phosphate backbone or as intermediates in the base-excision repair (BER) pathway (Aubuchon and Verma, 2024; Lindahl, 1993; Lindahl, 2004). Abasic lesions pose a significant threat to genome integrity because they lack the critical base-pairing information required for faithful DNA replication and transcription, resulting in the blockage of DNA (Hogg et al, 2004) and RNA polymerases (Wang et al, 2018; Yu et al, 2003). AP sites represent an intrinsic vulnerability in DNA due to their inherent fragility, which can lead to spontaneous strand breakage (Lindahl, 2016). Abasic DNA breakage can be also caused by nucleases and lyases that attack either the phosphodiester backbone or the aldehyde moiety left behind on the deoxyribose following base loss (Lindahl, 1993, 2004).

Although several mechanisms are involved in AP site tolerance (Thompson and Cortez, 2020), RAD51 had not been directly implicated in AP site metabolism until recent work highlighted its role in binding and protecting AP sites from nuclease attack during DNA replication. A multidisciplinary approach including cryo-electron microscopy (cryo-EM) demonstrated that RAD51 nucleofilaments specifically recognize and bind to AP sites in both single- and double-stranded (ds) DNA (Hanthi et al, 2024). Cryo-EM highlighted the ability of RAD51 nucleoprotein filaments to interact with DNA through its ATPase domain, where the DNA-binding loops L1 and L2 play a key role. These loops normally facilitate the formation of RAD51 filaments by stabilizing the DNA backbone and promoting base unstacking, allowing RAD51 to align nucleotides into triplet motifs (Appleby et al, 2023a; Xu et al, 2017). In the presence of an AP site, the absence of a nucleotide base creates a structural gap, which can be appreciated in a dsDNA

molecule containing an abasic strand with regularly spaced AP sites annealed to an undamaged complementary one (Fig. 2A). Cryo-EM structures reveal that the RAD51 nucleofilament preferentially associates with such lesions, positioning RAD51's L1 and L2 loop domains directly over the AP site (Hanthi et al, 2024). This allows the protein to anchor itself within the damaged DNA region without disrupting the integrity of the filament. The valine residue at position 273 (V273) in loop L2 partially occupies the space left by the missing base, stabilizing the interaction (Fig. 2B,C). Moreover, kinetic studies using surface plasmon resonance indicate that the association rate of RAD51 with DNA increases as the number of AP sites within the sequence rises, further supporting the hypothesis that RAD51 filaments preferentially nucleate at these sites (Hanthi et al, 2024). Such mechanism is likely to be conserved throughout evolution and might have been the primordial function of RAD51 orthologs such as RecA in more ancient species, allowing to protect DNA structure in different environments that promote AP site formation (Brendel et al, 1997).

Remarkably, the cryo-EM structure of RAD51 nucleofilaments bound to dsDNA containing AP sites only on one side revealed RAD51 preferential association with the strand carrying the lesions (Hanthi et al, 2024) (Fig. 2B). This selective strand discrimination suggests that RAD51 recognizes and protects abasic DNA in a highly specific manner. Intriguingly, RAD51 paralogs have been shown to preferentially interact with DNA containing an AP site and to promote AP site tolerance, implying that AP site binding capability is a broad feature of RAD51-like proteins (Rosenbaum et al, 2019).

However, direct RAD51 binding of AP sites does not directly explain the accumulation of AP sites and ssDNA gaps observed in HR-defective cells (Hanthi et al, 2024). AP lesions are typically repaired by BER, but if left unrepaired, they can persist in DNA templates during replication. Surprisingly, the absence of RAD51 or BRCA2 leads to a significant increase in AP sites (Hanthi et al, 2024). Unrepaired AP sites promote the accumulation of replication-associated ssDNA gaps acting as replication obstacles, stalling DNA polymerases and triggering DNA-synthesis re-priming via PrimPol (Garcia-Gomez et al, 2013; Mouron et al, 2013; Quinet et al, 2020) (Fig. 1B).

Among the mechanisms preventing the accumulation of AP sites in replicating DNA templates, there could be the ability of RAD51 to bind ssDNA generated by the unwinding of replicating templates, shielding it from access by DNA glycosylases. In HR-deficient cells, the exposed ssDNA formed during DNA replication becomes a substrate for single-strand-selective monofunctional uracil DNA glycosylase (SMUG1), an enzyme responsible for removing uracil and oxidized or deaminated bases, such as 5-hydroxymethyl-uracil (5hmU). This enzymatic activity leads to the widespread formation of AP sites in the DNA template, resulting in the accumulation of ssDNA gaps (Hanthi et al, 2024) (Fig. 2D). Inactivation of DNPH1, an enzyme that eliminates cytotoxic nucleotide 5-hydroxymethyl-deoxyuridine (5hmdU) monophosphate and thereby prevents its incorporation in genomic DNA, has been found to potentiate PARP1 inhibitor sensitivity of BRCA-deficient cells, demonstrating the toxicity of 5hmU (Fugger et al, 2021).

RAD51 may also limit SMUG1 activity on dsDNA ahead of replication forks, since the SMUG1 catalytic domain can interact with both strands at the damaged base (Pettersen et al, 2007).

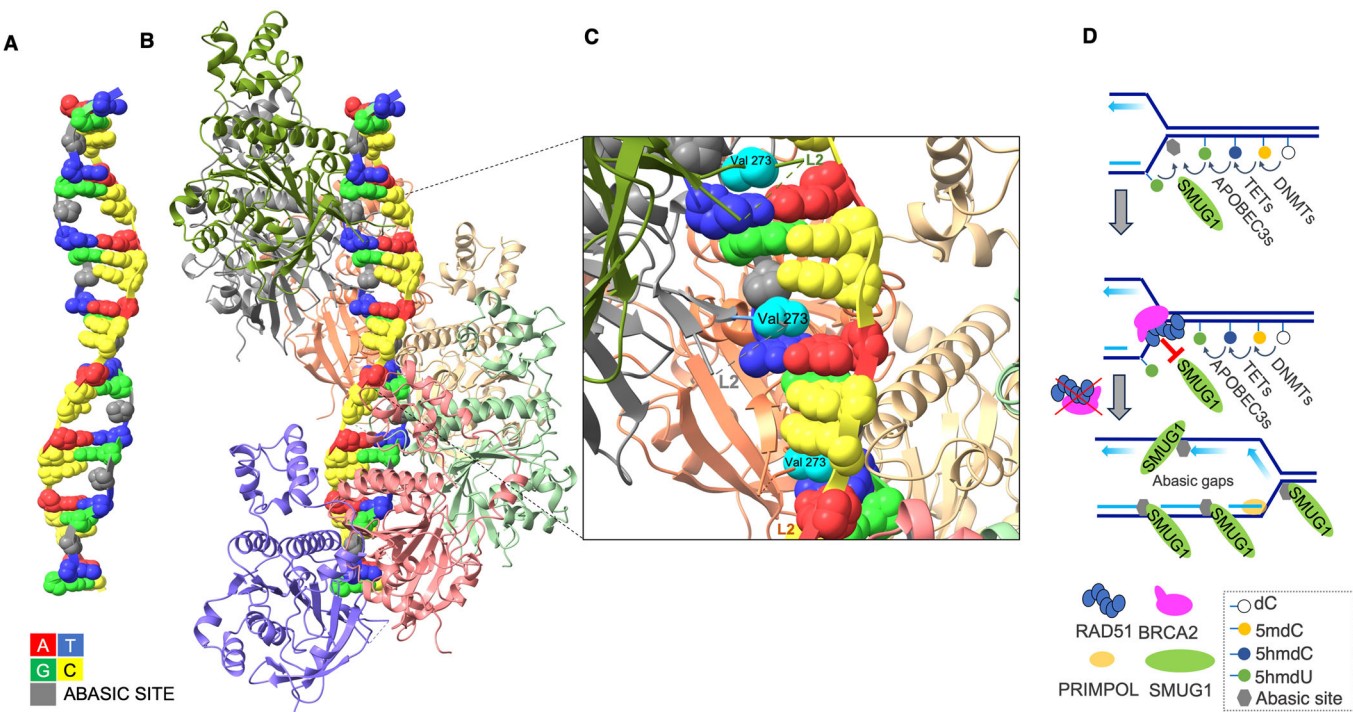

**Figure 2.  Structure–function of RAD51 on abasic DNA and sources of AP sites at replication forks.**

(**A**) Cryo-EM reconstruction of a DNA duplex composed of one undamaged strand and a complementary strand carrying an abasic (AP) site (gray) every third nucleotide, obtained by image averaging. The DNA sequence is indicated by the color key. (**B**) Cryo-EM view of a RAD51 nucleofilament comprising six protomers assembled on the AP-containing duplex shown in (**A**). (**C**) Close-up of the AP-site recognition pocket: RAD51 Val273 within the L2 loop inserts into the void created by the missing base. All views were generated from the 8RFC structure deposited in RCSB protein data bank (Hanthi et al, 2024). (**D**) Schematic of AP-site generation at forks: DNMT-dependent cytosine methylation yields 5-methylcytosine (5mC); TET enzymes oxidize 5mC to 5-hydroxymethylcytosine (5hmC); APOBEC3 deaminates 5hmC to 5-hydroxymethyluracil (5hmU); SMUG1 excises 5hmU to generate AP sites, which can accumulate with SMUG1 bound.

However, in vitro or when overexpressed, as in BRCA-defective cells, SMUG1 is capable of excising uracil from ssDNA, suggesting that it could target ssDNA exposed during DNA replication (Hanthi et al, 2024; Haushalter et al, 1999; Taglialatela et al, 2021). RAD51 may function by continuously trailing replication forks in a dynamic, treadmill-like mechanism, wherein short RAD51 nucleofilaments repeatedly assemble and disassemble on DNA, preventing its exposure to enzymes like SMUG1 (Fig. 2D). This process might contribute to explain the accumulation of AP sites in the absence of BRCA2 (Hanthi et al, 2024). The ability to recognize AP sites might render this action more effective in regions where some AP lesions are already present and where they tend to accumulate in clusters. How RAD51-mediated protection complements the function of other AP site binding proteins, including HMCES (Mohni et al, 2019), remains to be determined.

Persistent DNA binding and slow release of SMUG1 following AP-site formation can itself contribute to DNA-polymerase stalling during replication (Musiani et al, 2025) (Fig. 2D). In line with these findings, SMUG1 silencing is sufficient to prevent the accrual of abasic ssDNA gaps during DNA replication in HR-defective cells (Hanthi et al, 2024; Taglialatela et al, 2021), and to rescue hmdU toxicity (Fugger et al, 2021).

RAD51 may also facilitate the direct repair of AP sites within gaps by re-annealing the parental templates. In this way, BER could act by filling in the missing base, using the complementary parental strand as a template to insert the appropriate nucleotide. This strand-exchange-assisted BER might promote the accurate repair of AP sites present in replicating templates. The re-annealing of gapped template strands with the complementary parental strand would lead to the formation of a D-loop with the sister duplex, in which the newly synthesized sister strand is displaced. This configuration is supported by direct EM observation of the repair of gapped lesions induced by alkylating agents in yeast (Giannattasio et al, 2014). This would allow DNA synthesis to resume following the repair and the elimination of the AP sites without causing nucleotide misincorporation, effectively coupling typical HR-mediated reactions with BER. This model is however difficult to reconcile with the finding that AP sites impair DNA strand exchange and D-loop formation via destabilization of a donor/acceptor pairing mechanism mediated by base triplet stepping (Lee et al, 2015). An alternative possibility is that the portion of the invading strand carrying one or more AP sites is excluded from the D-loop, generating an abasic segment that is subsequently cleaved by the lyase activity of MRE11 (Hanthi et al, 2024; Larson et al, 2005).

Intriguingly, separation of function has been shown for BRCA2 alleles that retain canonical HR-mediated DSB repair, yet fail to suppress replication-coupled lesions. In BRCA2, substitutions in the conserved C-terminal RAD51-stabilizing region, including serine at codon 3291 (S3291), support efficient HR but permit

persistence of replication-associated gaps, suggesting that gap suppression is genetically separable from DSB repair (Lim et al, 2024). On the other hand, mutations that selectively disrupt HR while leaving replication-gap suppression intact have not been identified so far. Also, it remains unclear if the analyzed mutants represent true separation-of-function or mere hypomorphic alleles, as suggested by the residual activity of the corresponding mutant peptides in stabilizing RAD51 nucleofilaments (Appleby et al, 2023b). Human and mouse cells expressing BRCA2 mutants that are defective in fork protection and gap suppression but retain HR proficiency can still prevent chromosomal aberrations (Lim et al, 2024). However, it is not known if they suppress the appearance of mutation signatures for base substitutions. These observations warrant additional studies to establish whether the gap-suppression function of HR proteins is as critical as their role in strand exchange–mediated DSB repair in promoting genome stability.

Taken together, the above findings establish a key function for HR proteins in preventing replication fork instability, possibly caused by AP site-induced lesions. This role is distinct from their well-characterized role in HR-mediated DSB repair, emphasizing a direct involvement in preventing ssDNA gaps and base lesions formation. This has important implications for understanding replication stress occurrence in HR-deficient tumors, and provides a rationale for therapeutic strategies aimed at targeting ssDNA gap accumulation and replication fork stability in cancer cells.

## Gap sealing

In response to HR deficiency, cancer cells frequently resort to translesion synthesis (TLS) as a compensatory mechanism to tolerate DNA damage. TLS involves specialized polymerases, including POLη, POLκ, and POLζ, which allow DNA synthesis across damaged bases by incorporating nucleotides opposite lesions (Anand et al, 2023; Zamborszky et al, 2017). This processing is spatially and temporally segregated (Tirman et al, 2021; Wong et al, 2020). In addition to deletions with microhomology, mutational signatures (such as SBS3 and SBS8) that involve C > T transitions and T > A transversions are strongly associated with HR-deficient cancers (Heeke et al, 2018; Polak et al, 2017). These mutational patterns reflect the underlying error-prone TLS repair mechanisms that compensate HR defects, and provide insight into the molecular processes shaping the tumor genome. Moreover, such signatures can be used to identify HR deficiency in tumors lacking obvious BRCA1/2 mutations, broadening the scope of HR-related cancer diagnostics and treatment strategies. The increased mutation rate in HR-deficient cells has significant clinical implications.

One of the TLS polymerases contributing to such mutational signatures is POLζ (Chen et al, 2022), which can fill ssDNA gaps generated in the absence of BRCA1 (Taglialatela et al, 2021). Another gap-filling polymerase involved in maintaining genome stability is POLθ (Belan et al, 2022; Mann et al, 2022; Schrempf et al, 2022), which has a major role in the repair of DSBs through MMEJ, an alternative repair pathway utilized in mitosis as a last resort before cell death (Brambati et al, 2023; Gelot et al, 2023; Mateos-Gomez et al, 2015). Its ability to fill replication gaps might even prevent DSB formation at replication forks. While the gap-filling function of POLθ has been predominantly observed on the lagging strand (Mann et al, 2022; Schrempf et al, 2022), its role on the leading strand cannot be excluded, especially in the presence of

AP sites. The unique ability to bypass AP sites, which POLθ shares with POLζ, may contribute to filling abasic DNA gaps that occur in the absence of BRCA2/RAD51 (Hogg et al, 2011). The dual role of POLθ in replication-gap filling and DSB repair underscores its significance in maintaining cell viability under conditions of HR deficiency. However, although these mechanisms enable replication to proceed, their error-prone nature introduces base substitutions and small insertions or deletions, contributing to the mutational signatures observed in HR-deficient tumors (Wood and Doublie, 2016).

## Detecting ssDNA gaps

ssDNA gaps associated with AP sites have been identified via an assay based on APE1 nuclease, which specifically detects AP sites within replicating DNA fibers (Hanthi et al, 2024). APE1 can cut AP sites within ssDNA gaps formed in nascent DNA fibers, leading to the shortening of labeled fibers and thereby revealing the presence of abasic ssDNA gaps. As similar levels of ssDNA gaps can be detected by either APE1 or S1 nuclease, these results indicated that most of the gaps arising in HR-defective cells contain AP sites. Inhibition of APE1 activity or genetic inactivation of APEX1 gene can cause DNA fiber shortening due to AP site accumulation, confirming the assay's utility in assessing the impact of AP sites on replication dynamics, and AP sites can be directly detected by labeling with the cell-permeable alkoxyamine probe AA3 (Hanthi et al, 2024).

Regarding the localization of DNA gaps, different techniques vary in their ability to detect them based on distance and the timing of their origin. Gaps within 2-3 kb distance from the fork junction can be easily visualized using DNA-EM. In contrast, DNA fiber assays can detect gaps over longer distances, typically ranging from tens to hundreds of kilobases from the fork junction.

Although DNA fiber spreading combined with nuclease treatment before cell collection is a powerful method for detecting ssDNA gaps (Meroni et al, 2023; Quinet et al, 2017), the ideal technique to assess the presence of gaps on just one of the replicated sister chromatids is based on the combing protocol, which physically separates sister DNA molecules by including proteinase treatment of the replicated nuclei. On the other hand, in the original DNA fiber spreading assay, sister DNA molecules remain held together by cohesin and other chromatin proteins, and therefore must both be gapped in close proximity for the assay to reveal fiber shortening (Meroni et al, 2023; Quinet et al, 2017). Interpretation of strand-specific gap localization according to this method requires caution, as variations in DNA fiber stretching protocols, timing of gap formation, sealing by alternative polymerases, and cell-type-specific differences in cohesin retention can all influence the interpretation of leading- versus lagging-strand gap patterns. Importantly, DNA fiber spreading under harsher conditions, which include cell collection by trypsinization followed by cell permeabilization and incubation in suspension with S1 or APE1 nucleases, achieves better strand separation and nuclease penetration, revealing sister-specific DNA gaps even without the need to perform combing (Hanthi et al, 2024; Mann et al, 2022; Ramakrishnan et al, 2024). Here, ssDNA gaps can be sensitively detected in HR- and BER-defective cells in the absence of any exogenous DNA-damaging agents, as confirmed by AI-based automatic scoring of DNA fiber lengths following S1 nuclease

treatment (Fagherazzi et al, 2025). Under such conditions, silencing of PrimPol expression does not fully eliminate all detectable gaps after POLθ inhibition in BRCA2-defective cells, indicating that POLθ plays a compensatory role in filling gaps that arise independently of PrimPol, potentially on the lagging strand (Mann et al, 2022).

Gap filling can be highly dynamic and immediately follow the passage of the fork, as shown by the observation that only the most proximal DNA-labeled track is subjected to nuclease digestion, whereas the most distal segments remain refractory to nuclease-mediated shortening (Hanthi et al, 2024). The occurrence of gaps located far from fork junctions, although not detectable by fiber assays or DNA-EM, can be inferred by widespread mutagenic signatures, revealing base changes generated by TLS-mediated filling of ssDNA templates that cannot be repaired by TS (Chen et al, 2022). It is worth mentioning that not all ssDNA gaps are equal: gaps induced by PARP1 inhibitors or DNA-damaging agents like camptothecin (CPT) persist for longer, as they can be detected during post-replicative repair (Seppa et al, 2025).

Overall, the described experimental approaches have revealed that ssDNA gaps, especially the ones associated with AP sites, are common in HR- or BER-defective cells, as validated by APE1- and S1-coupled fiber assays, direct AP labeling, and SMUG1 and APEX1 perturbations. While EM maps can be used to fork-proximal gaps, stringent fiber protocols can capture longer-range and sister-specific gaps. As re-priming of DNA synthesis stalled by AP sites and POLζ- and POLθ-mediated filling creates measurable, potentially targetable dependencies, the development of new techniques allowing practical and reliable detection of AP sites and abasic gaps in human tissues will be important for clinical applications. Possible future applications warranting further development include nanopore-mediated single-molecule sequencing, in which long-read signals can register current AP-site-associated disruptions and map AP sites and gap landscapes at scale, directly from native DNA (Muller et al, 2019).

## Abasic ssDNA gaps originating from epigenetic modification

A common source of transient AP sites in DNA in higher eukaryotes is DNA methylation metabolism. DNA methylation is primarily maintained by DNMT1, which ensures the inheritance of methylation patterns across replication, while DNMT3B is responsible for de novo methylation (Prasad et al, 2021). Targeted experiments using an auxin-inducible degron (AID) system to selectively degrade DNMT1 in human cells revealed that a reduction in about 50% global 5mC levels (Scelfo et al, 2024b) decreases AP site formation and suppresses ssDNA gaps in BRCA2-deficient cells (Hanthi et al, 2024). Once a cytosine is methylated, it can undergo further oxidation by TET enzymes, producing intermediates such as 5-hydroxymethylcytosine (5hmdC) (Fig. 2D) and others, including 5-formylcytosine (5fdC), and 5-carboxylcytosine (5cadC) (Pfaffeneder et al, 2014). Depletion of TET2 leads to a marked reduction in abasic ssDNA gaps, supporting the hypothesis that the oxidation of methylated cytosines contributes significantly to the generation of replication-associated AP sites (Hanthi et al, 2024). Knockdown of APOBEC3B partially decreases accumulation of abasic gaps, suggesting that 5hmdC deamination to 5hmdU (Pecori et al, 2022) contributes to

abasic site formation following SMUG1-mediated removal of 5hmU (Hanthi et al, 2024) (Fig. 2D). A direct conversion of thymidine into 5hmdU by TET2 is also compatible with the main requirement of 5mC in formation of ssDNA gaps, as 5mdC might be required to anchor TET2 on DNA and to promote transformation of adjacent T into 5hmdU (Pfaffeneder et al, 2014). Bases modified by TET enzymes also serve as substrates for thymine DNA glycosylase (TDG), which excises them in the context of duplex DNA, leaving behind an abasic site. If these lesions are not promptly repaired through BER, they may contribute to the formation of ssDNA gaps during replication (Wang et al, 2022a). However, silencing of TDG does not lead to suppression of ssDNA gaps, ruling out regular DNA methylation turnover in gap accumulation (Hanthi et al, 2024).

These findings highlight a mechanistic pathway in which DNA methylation and its subsequent modifications generate AP sites that, if left unrepaired, accumulate in replicating DNA and contribute to genomic instability. This concept is further supported by the discovery of APE1-mediated degradation of 5hmC-marked stalled replication forks (Kharat et al, 2020). The interplay between epigenetic modifications and DNA damage suggests that AP site accumulation is not simply a passive consequence of DNA metabolism, but a process driven by enzymatic activity. This connection is particularly relevant in HR-deficient cells, such as those lacking BRCA2, where the inability to efficiently repair AP sites exacerbates replication stress and increases vulnerability to genomic instability. Again, this function help to explain the essential role of RAD51 in higher eukaryotes, further emphasizing the complex relationship between epigenetic regulation and DNA damage repair mechanisms. The link between DNA methylation and DNA repair is further highlighted by the concurrent evolution of DNA methyltransferase activities and BRCA genes (Rosic et al, 2018).

Loss of DNA methylation, a phenomenon frequently observed in various human diseases, has the potential to influence the formation and repair of AP sites. Cancers often show global hypomethylation, which has been associated with genomic instability and altered gene expression (Yamada et al, 2005). Although hypomethylation is typically linked to genomic instability through oncogene activation or loss of imprinting (Ehrlich, 2009; Holm et al, 2005; Sheaffer et al, 2016), it may at the same time alleviate one source of DNA lesions, by restricting AP site formation: fewer methylated cytosines means fewer substrates available for TET-mediated oxidation, reduced production of oxidized intermediates such as 5hmdU, and less subsequent excision events that would result in AP sites and ssDNA gaps (Fig. 2D). This could represent a compensatory mechanism that helps stabilize the genome in response to other oncogenic stresses, such as increased replication demand, oxidative DNA damage and lack of effective DNA repair. Such reduction in damage might improve the fitness of BRCA1/BRCA2-defective cells by lowering the burden of replication-blocking AP lesions linked to DNA-methylation processing. Accordingly, global DNA hypomethylation at CpG islands coupled with local hypermethylation is a hallmark of breast cancer (Brinkman et al, 2019). Impaired BRCA1 function has also been directly linked to global DNA hypomethylation and loss of genomic imprinting, causing oncogene activation and cancer predisposition due to a lack of BRCA1-dependent regulation of DNMT1 (Shukla et al, 2010). In contrast, direct connections remain to be established for BRCA2, but recent methylome profiling of

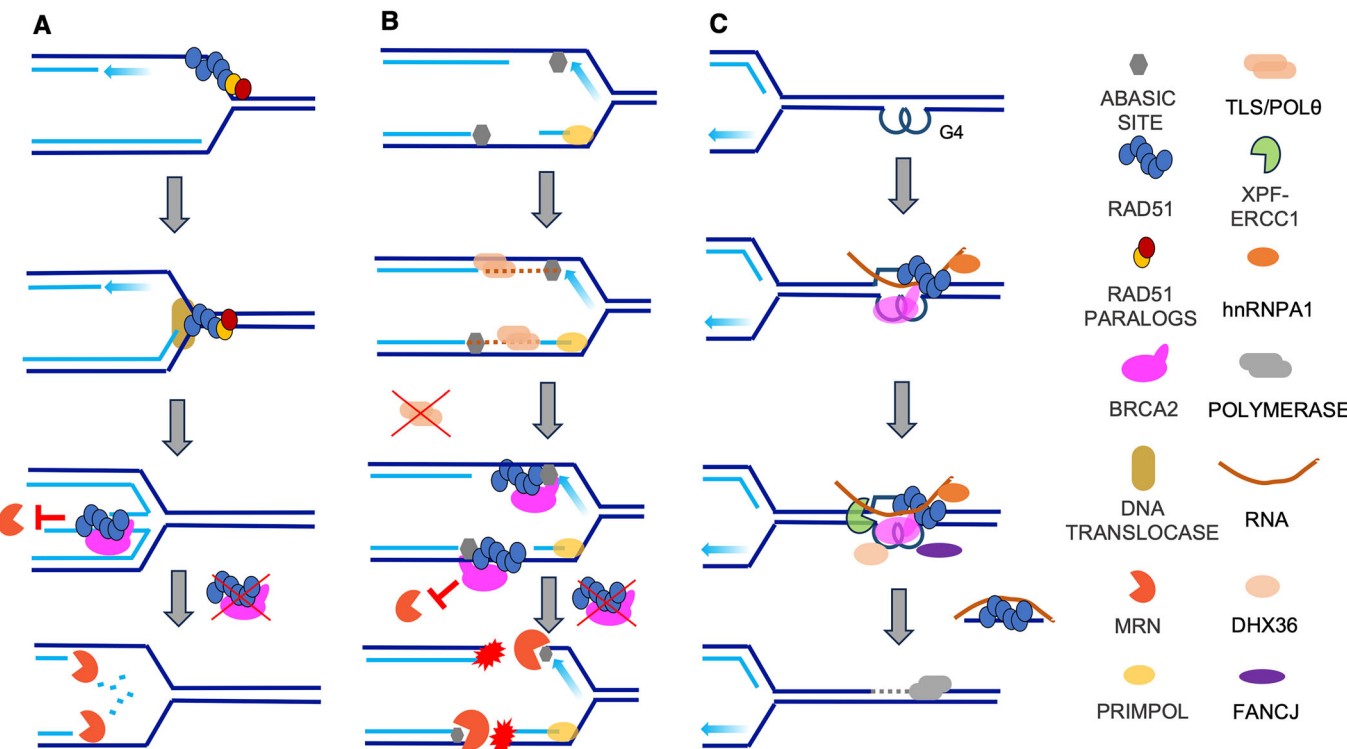

**Figure 3. HR-independent roles of RAD51 in protecting DNA replication intermediates and structured DNA.**

(A) RAD51, together with its paralogs, initiates fork reversal, which is then extended by DNA translocases. BRCA2-stabilized RAD51 filaments shield the reversed arms from degradation by the MRN complex and other nucleases. (B) AP sites stall DNA synthesis. Re-priming occurs on the lagging strand by replicative polymerases and on the leading strand by PrimPol, generating ssDNA gaps that are partially filled by TLS and POLθ. RAD51 filaments preferentially nucleate at AP sites to protect AP-containing gaps from MRN-dependent endonucleolytic cleavage and to prevent fork breakage. (C) In non-replicating DNA, BRCA2–RAD51 promotes formation of RNA-DNA hybrids on the displaced strand to assist G-quadruplex (G4) resolution, in concert with DHX36 and FANCJ helicases. Targeted cleavage adjacent to the hybrid by XPF-ERCC1 permits gap filling and completes G4 removal, thereby preventing stalling of subsequent replication forks.

circulating tumor DNA from BRCA2-mutant breast cancers reveals hypomethylation across DNA-repair and cell-cycle genes, consistent with a broader hypomethylated state in BRCA-mutant tumors (Grisolia et al, 2024). Hypomethylation might also favor uncontrolled hyper-recombination of repetitive sequences, as in the case of centromeres lacking the repeat-element-protecting factors BRCA2 or PALB2 (Graham et al, 2025). These combined observations support the hypothesis that hypomethylation may relieve AP-site-driven replication stress and confer a selective growth advantage in BRCA-defective contexts, synergizing with oncogenic chromosome rearrangements.

## Protection of vulnerable DNA structures by RAD51 and other HR proteins

### BRCA- and RAD51-mediated fork protection

Beyond preventing the occurrence of spontaneous lesions, RAD51 plays a critical role in protecting DNA from degradation. Early work demonstrated that ssDNA gaps accumulating behind replication forks in the absence of RAD51 are extended by MRE11-dependent degradation of nascent DNA. Inhibition of MRE11 nuclease activity suppresses this degradation, highlighting

the importance of RAD51 in protecting nascent DNA and ensuring continuous DNA synthesis (Hashimoto et al, 2010).

Expanding on these findings, an assay based on progressive shortening of labeled DNA fibers upon fork stalling revealed that RAD51, in conjunction with BRCA2, stabilizes stalled replication forks by preventing extended MRE11-dependent degradation of nascent DNA (Schlacher et al, 2011). This protective role of RAD51 nucleofilaments is distinct from their function in HR-mediated DSB repair and acts by counteracting key nucleases, including MRE11, EXO1, DNA2, SLX1-4, MUS81, EEPD1 and others involved in the degradation of unprotected stalled replication forks (Halder et al, 2022; Kolinjivadi et al, 2017a; Kolinjivadi et al, 2017b; Longo et al, 2025; Mijic et al, 2017; Schlacher et al, 2011; Seppa et al, 2025; Zadorozhny et al, 2017; Zellweger et al, 2015). Thus, degradation of nascent DNA strands has rapidly emerged as a hallmark of HR deficiency and plays a central role in the genomic instability observed in HR-deficient tumors.

The major trigger for extensive nascent DNA degradation are reversed replication forks (Kolinjivadi et al, 2017a; Kolinjivadi et al, 2017b; Mijic et al, 2017; Taglialatela et al, 2021; Zellweger et al, 2015). Together with RAD51 paralogs, RAD51 might initiate fork reversal by facilitating re-annealing of unwound templates and the disengagement of nascent strands, a process that is then completed by translocases such as SMARCAL1, HLTF, ZRANB3 and FBH1

(Adolph and Cortez, 2024; Berti et al, 2020; Liu et al, 2023) (Fig. 3A). BRCA2 stabilizes binding of RAD51 nucleofilaments to single- or double-stranded reversed branches, thereby preventing degradation of newly synthesized DNA strands and preserving replication fork stability (Fig. 3A). This pathway is possibly triggered by impaired DNA damage surveillance mechanisms, which affect fork reversal (Sogo et al, 2002). While RAD51's role in protecting nascent DNA is well established, its contribution to fork reversal is less clearly defined at the molecular and structural levels. It is tempting to speculate that by shielding base-damaged DNA from endonucleolytic cleavage, RAD51 may prevent the dissipation of positive supercoiling ahead of replication forks, which is required for efficient fork reversal (Atkinson and McGlynn, 2009; Postow et al, 2001). According to this untested model, RAD51 might indirectly contribute to fork reversal via DNA protection.

In the absence of RAD51-mediated fork remodeling, ssDNA gaps behind forks could alternatively be formed on replication intermediates due to fork reversal impairment, which favors nascent DNA synthesis re-priming (Quinet et al, 2020). Clear separation-of-function mutations would be valuable for disentangling gap suppression, fork reversal, DNA protection, and HR-mediated DSB repair, and for further strengthening these non-canonical roles of HR proteins. Additional characterization of available RAD51 mutants impairing fork protection and strand invasion could help to better clarify the different functions of RAD51 (Zadorozhny et al, 2017; Joudeh et al, 2025). The RAD51-II3A mutant containing R130, R303, and K313 changed to alanines has been useful for demonstrating that fork reversal and protection from MRE11 degradation do not require RAD51's strand exchange activity (Mason et al, 2019). The RAD51 S181P mutation, which selectively disrupts the association with the BRCA2 C-terminus while maintaining interaction with the BRC repeat, was found to be defective for fork protection and restart (Son et al, 2024). Finally, other factors controlling RAD51 filaments dynamics, such as RADX, could be important for regulating DNA replication fork remodeling and stability (Adolph et al, 2021; Bhat et al, 2018; Krishnamoorthy et al, 2021).

Crucial for these regulatory aspects is the modular structure of the BRCA2 protein, whose various domains differentially regulate RAD51 functions. While the BRCA2 segment containing RAD51-binding BRC repeat domains is critical for HR-mediated DSB repair, a short C-terminal peptide binds and stabilizes RAD51 nucleofilaments on ssDNA. This C-terminal region is disproportionately important at stalled or reversed forks, where stable RAD51 filaments shield nascent DNA from nuclease-mediated digestion and preserve fork restart competence. Mutations in this segment of BRCA2 can leave HR-mediated DSB repair largely intact, while at the same time compromising fork protection (Longo et al, 2025; Schlacher et al, 2011).

BRCA1 also participates in fork protection. PIN1-catalyzed prolyl isomerization of BRCA1 in complex with BARD1 promotes RAD51 loading at stalled forks independently of the BRCA1-PALB2 axis required for HR-dependent DSB repair. Engineered mutants affecting BRCA1 prolyl isomerization separate fork protection from HR activity, supporting a distinct, replication-fork protection function for BRCA1 (Daza-Martin et al, 2019). Likewise, the BRCA1-BARD1 complex, which normally promotes resection, switches to a DNA nuclease-protection function in the presence of RAD51 (Ceppi et al, 2024).

RPA, the other key ssDNA-binding protein, remains abundant after RAD51 removal, but unlike RAD51 is not able to protect DNA intermediates from degradation in vivo. This may be explained by the fact that in RAD51 nucleofilaments, ssDNA lies inside a highly stable protein structure (Appleby et al, 2023a; Davies and Pellegrini, 2007; Hanthi et al, 2024; Pellegrini et al, 2002); while it dynamically engages with and partially wraps around RPA heterotrimers, leaving exposed DNA accessible to enzymes such as DNA polymerases and nucleases (Yates et al, 2018; Chadda et al, 2024). Still, exhaustion of all available RPA complexes has been found to induce DSB formation (Toledo et al, 2017), but this appears to be a late event in the processing of DNA replication forks triggered in the absence of functioning DNA damage and replication checkpoints (Bertolin et al, 2025). Intriguingly, RPA and RAD51 seem to coexist on ssDNA, where they occupy different areas of the DNA. In particular, ssDNA regions of at least 18 consecutive nucleotides between neighboring RPA molecules promote RAD51 assembly onto RPA-coated ssDNA (Ding et al, 2023). This reaction is aided by RAD52, which by increasing the spacing between RPA molecules uncovers ssDNA for RAD51 loading, prior to further stabilization via BRCA2 (Appleby et al, 2023b; Ding et al, 2023).

The accumulation of AP sites in the absence of BRCA2/RAD51 presents a significant challenge to the stability of replication forks. The persistent presence of these lesions can generate abasic ssDNA gaps that render replication forks vulnerable to nuclease-mediated processing, as shown by the ability of MRE11 (Hanthi et al, 2024; Larson et al, 2005) and APE1 (Hoitsma et al, 2023) to cleave ssDNA at abasic gaps within DNA replication forks. These regions are initially stabilized by RAD51 nucleofilaments, which preferentially bind to AP sites and prevent their endonucleolytic cleavage, and further protected by BRCA2, which stabilizes RAD51 nucleofilaments on AP site-containing DNA (Hanthi et al, 2024) (Fig. 3B). EM analysis of replication forks from HR-deficient *Xenopus laevis* egg extracts and human cells has revealed broken forks, recognizable by asymmetric Y-shaped structures and discontinuous DNA tracts (Hanthi et al, 2024; Mann et al, 2022; Pena-Gomez et al, 2025). In the absence of RAD51, these structures likely arise from stalled replication forks that cannot be stabilized upon encountering abasic DNA (Fig. 3B). Without this protective mechanism, replication forks undergo active breakage as the exposed ssDNA regions containing AP sites become susceptible to MRE11-RAD50-NBS1 (MRN) complex-dependent cleavage (Fig. 3B). Through this mechanism, RAD51 helps prevent replication fork collapse and possibly facilitates fork restart, ensuring continued DNA synthesis even in the presence of lesions. Active cleavage and fork collapse occur independently of cell-cycle progression, as shown by the surge of broken intermediates detected by DNA-EM in *Xenopus laevis* egg extracts arrested in S phase, without entry into mitosis or subsequent cycles (Hanthi et al, 2024).

Interestingly, while APE1, the primary AP endonuclease, has limited activity on ssDNA at physiological concentrations, the MRN complex exhibits lyase activity capable of cleaving natural AP sites (Hanthi et al, 2024; Larson et al, 2005). While this cleavage may not involve endonuclease activity, as MRE11 is unable to efficiently process undamaged ssDNA (Seppa et al, 2025), it is blocked by the MRE11 endonuclease inhibitor PFM01 (Shibata et al, 2014), which also prevents DSB formation in HR-deficient cells. The N-terminal lyase activity of MRE11 (Larson et al, 2005)

may be responsible for severing ssDNA gaps containing AP sites and could be indirectly influenced by the interaction of PFM01 with MRE11, and CtIP and NBS1 may be involved in promoting such cleavage in vivo, despite being unable to stimulate the processing of ssDNA gaps in vitro on AP-site-free substrates (Mann et al, 2022; Schrempf et al, 2022; Seppa et al, 2025). Alternatively, other nucleases capable of processing forks activated by the MRN complex and inhibited by PFM01 might catalyze fork breakage in the presence of AP sites. Direct in vivo assessment is complicated by the fact that inhibition of MRE11 nuclease functions or complete removal of MRE11 is lethal, possibly due to other unknown essential roles of the MRN complex.

The vulnerability of forks stalled by unprotected AP sites is especially enhanced in the absence of effective TLS- and POLθ-mediated processing of abasic ssDNA gaps (Fig. 3B). As POLθ inhibition exacerbates fork cleavage (Mann et al, 2022; Schrempf et al, 2022), it is likely that POLθ participates in DNA synthesis across AP sites. Consistent with a complementary function of POLθ in filling AP sites containing ssDNA gaps in HR-defective cells, POLθ has been shown to have a role in bypassing abasic sites (Hogg et al, 2011; Yi et al, 2023). POLθ might also be involved in the repair of two dsDNA molecules flanking the gapped regions cleaved by the MRN complex in proximity to a protein adduct formed with AP sites (Chandramouly et al, 2021). Together, these findings suggest that it might be useful to therapeutically combine strategies leading to increased AP sites with inhibitors of POLθ. Notably, metabolic alterations triggered by free radicals inducing base damage, rather than fork stalling per se, have been proposed to compromise the stability of nascent DNA in cells deprived of HR-mediated fork protection (Somyajit et al, 2021). This would fit with a role for AP sites generated following oxidative stress-induced base damage in triggering nascent DNA degradation and fork collapse.

The DNA-protective function of RAD51 is highly conserved and might predate the evolution of strand exchange. Consistent with this, the DNA-binding domain of the bacterial RAD51 ortholog RecA is much more conserved throughout evolution than the N- and C-terminal regions necessary for strand exchange (Joudeh et al, 2025). Accordingly, in bacterial cells, RecA plays a direct role in protecting nascent DNA strands from degradation. RecA polymerizes on ssDNA and stabilizes replication forks stalled by DNA lesions, including AP sites, preventing extensive nucleolytic degradation (Ramirez-Otero and Costanzo, 2024). Studies in *E. coli* mutants lacking RecA demonstrated that replication intermediates are highly susceptible to degradation by nucleases such as RecBCD and UvrABC, underscoring the protective function of RecA beyond its role in strand exchange (Courcelle and Hanawalt, 2003).

The persistence of ssDNA gap regions in HR-defective cells, expanded by the MRN complex in conjunction with CtIP, EXO1, and DNA2, may also promote fork collapse when carried over into the subsequent S phase (Seppa et al, 2025). A similar effect may result from single-strand nicks in the template that remain unrepaired until the next S phase (Serrano-Benitez et al, 2023). Therefore, the persistence of unsealed DNA gaps or nicks in nascent or parental strands, resulting in strand-breakage when encountered by the fork in subsequent cell cycles, may constitute an additional mechanism of fork rupture. This might happen more often in conditions in which cells are treated with PARP inhibitors (Aubuchon and Verma, 2024; Seppa et al, 2025; Simoneau et al,

2021). In this case, the protective function of RAD51 might be due to its dual role in preventing excessive duplex DNA nicking and resection of ssDNA gaps.

The protective role of RAD51 at replication forks can be compromised if the protein becomes excessively engaged in unresolved recombination intermediates. Prolonged strand invasions or stalled D-loop structures may sequester RAD51, reducing its availability for fork stabilization. Certain genetic and environmental stressors, including oncogene activation, replication fork collisions, and high AP site accumulation, may contribute to this phenomenon by promoting excessive RAD51 binding to DNA. Consistent with this hypothesis, recent work shows that BRCA1-deficient cells accumulate AP sites and ssDNA gaps that trap RAD51, thereby preventing the assembly of RAD51 foci after ionizing radiation (Peng et al, 2025). Under normal conditions, helicases such as BLM and RECQ5 dismantle excessive RAD51 filaments to resolve recombination intermediates (Hu et al, 2007; Patel et al, 2017). However, when these helicases are defective, unresolved RAD51-bound recombination structures persist, further depleting the nuclear pool of RAD51. Aberrant cytoplasmic localization of RAD51, observed in certain cancer types, may further limit its availability for nuclear DNA repair functions and thus exacerbate replication stress (Wang et al, 2024).

In a similar way, the activity of Fidgetin-like 1 (FIGNL1) may influence RAD51 function in replication fork protection. FIGNL1 is a protein that belongs to the AAA+ ATPase protein family. It exhibits ATPase activity and interacts with RAD51, which facilitates the disassembly of nucleofilaments from HR intermediates and/or prevents their formation (Carver et al, 2025; Ito et al, 2023; Kumar, 2025; Zainu et al, 2024). On the one hand, while FIGNL1 serves to prevent toxic recombination events, excessive filament disassembly could inadvertently deplete RAD51 at replication forks, rendering stalled forks more susceptible to nucleolytic degradation. On the other hand, lack of FIGNL1 might result in RAD51 sequestration on dsDNA, limiting its availability in protecting ssDNA at forks. Dysregulation of FIGNL1 activity could therefore represent an additional mechanism by which replication fork instability arises in HR-deficient cells.

## DNA protection via Fanconi Anemia pathway proteins

Similar to RAD51, Fanconi anemia (FA) pathway proteins, including FANCD2 and FANCI, also play a role in suppressing DNA degradation (Schlacher et al, 2012) and in mitigating the effects of AP sites on DNA replication preventing fork collapse (Pena-Gomez et al, 2025). These proteins promote DNA interstrand cross-links and facilitate HR by stabilizing RAD51 nucleofilaments (Walden and Deans, 2014). The FANCD2–FANCI complex can adopt a closed conformation that clamps around dsDNA (Shakeel et al, 2019), allowing it to slide along it and monitor irregularities (Alcon et al, 2024). When encountering a double-to-single strand junction, which often signals the presence of DNA damage, the complex transitions into a locked state that is likely involved in initiating repair. Given that AP sites are frequent DNA lesions capable of inducing such junctions during DNA replication, the FANCD2–FANCI complex might play a role in mitigating their impact. Recent studies have shown that FANCD2- and FANCI-deficient cells exposed to 5hmC accumulate replication fork abnormalities leading to fork breakage, as shown by DNA-EM

(Pena-Gomez et al, 2025). If confirmed, this would implicate FANCD2-FANCI in protecting replication forks from AP-site-associated damage generated by SMUG1 processing of 5hmU, the deaminated derivative of 5hmC incorporated into nascent DNA. Whether these effects occur entirely independently of RAD51 remains to be established. Furthermore, BER may be initiated during S phase, acting on post-replicative segments of sister chromatids and indirectly affecting replication fork progression in the absence of FANCD2-FANCI. The differential capacities of FANCD2 and FANCI in recognizing and directly protecting double- versus single-stranded DNA with or without AP sites remain to be determined.

Intriguingly, replication fork disruption caused by 5hmC in FANCD2- or FANCI-deficient cells is alleviated by HMCES loss, in apparent contrast to the proposed AP-site-protective role of HMCES (Pena-Gomez et al, 2025). Reduced HMCES also promotes ssDNA gap-filling via TLS (Hanthi et al, 2024; Mehta et al, 2020). The unexpected HMCES toxicity at forks may be linked to its release mechanism from cross-links with AP sites (Donsbach et al, 2023): crosslink reversal dependent on DNA structure enables fork bypass without any problems, while in HR-deficient cells with excessive AP sites or 5hmC, persistent HMCES binding to abasic DNA may stall forks even further and potentiate damage.

## BRCA2, RAD51, and G quadruplexes

A wealth of experimental data suggests that BRCA2–RAD51 execute non-recombinational roles that concurrently control DNA G-quadruplexes (G4) and R-loop structures. Multiple orthogonal assays have found sharp rises in pan-nuclear DNA:RNA hybrid accumulation in both cycling and quiescent chromatin in BRCA2-depleted cells (Bhatia et al, 2014). Also, BRCA2 controls DNA:RNA hybrids by mediating RNAseH2 recruitment on damaged chromatin (D'Alessandro et al, 2018). A large fraction of chromatin-bound BRCA2 can be mapped to G4-forming DNA sequences, implicating G4 islands as major BRCA2 substrates in unperturbed cells (Wang et al, 2022b). Consistent with this, recent biochemical studies found that BRCA2 and RAD51 rapidly load onto model G4 structures, and that they are indispensable for the ensuing hnRNPA1- and RNA-guided G-loop assembly that precedes G4 unwinding (Fig. 3C). RAD51 filament formation nucleates G-loop assembly and at the same time licenses its orderly dismantling following unwinding of the G4 structure on the complementary strand by DHX36 and FANCJ helicases (Sato et al, 2025), possibly averting their global build-up and the consequent replication stress (Fig. 3C). By nucleating a RAD51 filament on the non-G4 strand, BRCA2 simultaneously stabilizes the G4/G-loop architecture. This configuration might shield AP sites that are intrinsically enriched in guanine-rich, oxidatively stressed DNA, until G4s are resolved. G4 structures are privileged hotspots for the formation of AP sites, with more than 60% of high-confidence AP sites found within 100 bp of a folded G4 (Roychoudhury et al, 2020) or a GC-rich promoter (Cai et al, 2022). G-rich runs in G4 quadruplexes and loops are easily converted into 8-oxoG DNA lesion. Excision of 8-oxoG by OGG1 within the conformationally restrained quadruplex leaves an AP site that is refractory to rapid APE1-mediated processing (Fleming et al, 2021), allowing the lesion to persist into S-phase, where it blocks replicative polymerases and triggers re-priming-dependent gaps. The

inherently high rate of spontaneous depurination at G-rich tracts further exacerbates this burden. Also, the G-loop cycle required to resolve G4s involves XPF-ERCC1 nuclease-mediated incision of the non-G4 strand, followed by DNA synthesis (Fig. 3C). If this step is delayed, transient single-stranded C-rich DNA opposite the G4 becomes a substrate for APOBEC3- or SMUG1-mediated base removal, seeding clusters of AP sites precisely where BRCA2–RAD51 filaments assemble.

An additional layer of regulation may stem from the oxidative turnover of 5mC. Since cytosine methylation modulates the thermal stability of G4s (Stevens et al, 2022), the DNA methylation landscape could fine-tune both the spatial distribution and urgency of BRCA2–RAD51-mediated protection. Highly methylated regions may foster thermodynamically stable G4 structures, enhancing BRCA2 recruitment and concurrently generating more AP sites requiring shielding. In contrast, DNA hypomethylation could lead to less stable G4s, reduced formation of abasic lesions, and a diminished reliance on the protective function of the BRCA2–RAD51 complex. In this light, safeguarding and repairing G4/G-loop structures while stabilizing AP sites might be two interlocking facets of a unified, preventive genome-maintenance pathway orchestrated by BRCA2–RAD51, a hypothesis to be experimentally explored.

# Therapeutic exploitation of abasic sites, ssDNA gaps, and fork collapse in HR-defective tumors

The accumulation of replication-associated ssDNA gaps and AP site-containing intermediates predisposing to fork collapse could be therapeutically exploited. The gap model places dsDNA disconti-nuities at the center of cytotoxicity of PARP1 inhibitors in BRCA-mutant cancers (Cong and Cantor, 2022). As lethality scales with the persistence of replication gaps, disabling pathways that promote clearance of AP sites or accelerate BER would be expected to intensify ssDNA gap burden, and to promote collapses that HR-defective cells cannot resolve or protect against. Such an increase in abasic gaps may be induced by inhibition of TLS polymerases and POLθ, which are able to fill these lesions by bypassing AP sites within ssDNA gaps, and POLθ inhibitors could indeed enhance fork collapse in HR-defective cells (Hanthi et al, 2024; Taglialatela et al, 2021). Inhibition of the lagging-strand maturation machinery through FEN1 inhibitors could also enhance the accumulation of unrepaired ssDNA gaps (Caldecott, 2024; Hanzlikova et al, 2018; Vaitsiankova et al, 2022). Alternatively, augmentation of AP sites and ssDNA gaps might be achieved by inhibiting factors promoting AP site repair. In particular, inhibition of ALC1/CHD1L, a poly(ADP-ribose)-dependent chromatin remodeler that facilitates access and progression of BER and SSB repair at abasic intermediates (Aubuchon and Verma, 2024; Ramakrishnan et al, 2024), might prolong the lifetime of AP-containing and SSB-linked intermediates, increase PARP1 retention, and heighten the prob-ability that reprimed gaps persist into S phase. In BRCA-mutant cells, where HR-mediated gap suppression is compromised, this should amplify the cytotoxic mechanism of PARP1 inhibitors by converting transient stalls into fork collapse and lethal double-strand breaks (Ortega et al, 2025). ALC1 inhibition could also reduce post-replicative cleanup of AP-derived gaps, further sensitizing HR-defective cells.

Importantly, in samples from patients that had progressed despite treatment with PARP inhibitors, the dominant resistance mechanism is restoration of BRCA1/2 function through reversion mutations, and longitudinal clinical profiling studies also showed BRCA1/2 reversion as the prevalent route of acquired resistance in the clinic. By contrast, the alternative route to resistance inferred from preclinical studies, including loss of TP53BP1 or other factors that relieve the end-resection block and partially restore HR-mediated DSB repair, appear uncommon in patient tumor samples (Harvey-Jones et al, 2024). Therefore, full restoration of all regulatory functions of BRCA1/2 proteins, including protection and prevention of AP sites and gaps, might be required to establish effective chemoresistance.

Another relevant mechanism of therapy resistance is the elevation of RAD51 protein levels. Increased RAD51 expression, often observed in tumor cells with p53 loss confers greater tolerance to DNA damage (Klein, 2008; Schild and Wiese, 2010). Similar findings are observed in cells that lack EMI1, a ubiquitin ligase adapter that restraining the cellular levels of RAD51 (Marzio et al, 2019). Paradoxically, although few studies have shown that high RAD51 levels partially rescue survival and HR in BRCA1-defective cells (Martin et al, 2007), in some instances high levels of RAD51 reduce the overall frequency of HR events that repair DSBs (Paffett et al, 2005).

In cancer cells overexpressing RAD51, replication forks are efficiently stabilized, preventing their collapse and reducing the efficacy of several chemotherapeutic agents, including platinum-based drugs, topoisomerase inhibitors, and PARP1 inhibitors (Klein, 2008; Marzio et al, 2019). Excess RAD51 might inhibit nucleolytic processing, thereby preserving fork integrity and allowing replication to resume after the resolution of replication stress. This enhanced fork protection mechanism could allow cancer cells to withstand significant DNA damage and continue proliferating, leading to treatment failure and tumor progression. The impact of RAD51 overexpression is particularly evident in aggressive cancers, including triple-negative breast cancer (Maacke et al, 2000; Martin et al, 2007), ovarian cancer (Hoppe et al, 2021), and glioblastomas (Morrison et al, 2021), where it correlates with poor prognosis and resistance to standard therapies. Targeting RAD51-mediated fork protection is therefore emerging as a promising strategy to counteract chemoresistance and enhance the efficacy of DNA-damaging treatments.

Targeting RAD51 might also help overcome the suppression of the cytoplasmic leakage of DNA fragments generated by MRE11 endonuclease activity, which trigger innate immunity pathways (Coquel et al, 2018; Jazayeri et al, 2008), including the ones controlled by cGAS STING, the activation of which correlates with strong anti-cancer immune responses (Wolf et al, 2016). One potential approach to target RAD51 protein is the use of proteolysis-targeting chimera (PROTAC) technology (Hinterndorfer et al, 2025), which offers a means of selectively degrading excess RAD51 in cancer cells. PROTAC molecules function as bifunctional agents, linking the target protein, in this case RAD51, to an E3 ubiquitin ligase, facilitating its ubiquitination and subsequent degradation by the proteasome. By reducing RAD51 levels, PROTAC-mediated degradation would weaken the ability of cancer cells to stabilize replication forks and repair DNA damage, thereby sensitizing them to chemotherapy. This strategy could be particularly effective in tumors with abnormally high RAD51

expression. Lowering RAD51 levels would make stalled replication forks more vulnerable to degradation by nucleases such as MRE11, EXO1, and DNA2, leading to fork collapse and increased DNA breakage, especially following inhibition of gap-filling activities of TLS polymerases. This approach could be combined with agents that induce AP sites or that promote the formation of nicks (Whalen et al, 2025). This would enhance the sensitivity of cancer cells to replication-stress-inducing agents and promote cell death, particularly in tumors already exhibiting defects in DNA repair pathways.

Beyond replication fork destabilization, PROTAC-mediated RAD51 depletion could also prevent resistance to PARP inhibitors that emerges through the restoration of HR and fork protection (D'Andrea, 2018). By limiting HR efficiency, PROTACs targeting RAD51 could restore the sensitivity of tumors to PARP inhibition, thereby extending the therapeutic window for patients with BRCA-proficient or HR-proficient cancers that would otherwise resist treatment. The development of selective small-molecule designed to target specific functional domains of RAD51, such as its DNA-binding region or its interaction interface with BRCA2, would be important to allow precise degradation while minimizing off-target effects. Future studies aimed at optimizing E3 ligase engagement and enhancing tumor-specific targeting will be critical to advancing this strategy into clinical applications.

## RAD51 as an anti-aging agent

Overexpression of a dominant-negative RAD51 variant reduces RAD51-mediated strand exchange, causing defective fork progression, premature ageing, and reduced tumorigenesis (Matos-Rodrigues et al, 2023). In a different direction, RAD51 over-expression itself slows DNA elongation and induces replication stress (Parplys et al, 2015). Unlike loss of HR mediators such as BRCA1/BRCA2, which increases cancer risk, these RAD51 perturbations underscore a narrow window of optimal RAD51 activity in which even modest sequence or dosage changes disrupt replication and compromise fitness and lifespan.

Notably, germline mutations near the ATPase and L1/L2 DNA-binding regions of RAD51, implicated in DNA and AP-site engagement, have been linked to congenital mirror movements (CMM) disorder (Franz et al, 2015). It is tempting to speculate that impaired RAD51 recognition or protection of spontaneous AP site-containing DNA contributes to ageing and neurodevelopmental pathology. AP sites, which increase with oxidative base damage and 5mC turnover (Aubuchon and Verma, 2024), block replication and transcription when unrepaired because DNA and RNA polymerases cannot traverse abasic lesions (Hogg et al, 2004; Wang et al, 2018; Yu et al, 2003). This mechanism, long implicated in age-related tissue dysfunction, merits renewed investigation in light of emerging non-canonical RAD51 functions at damaged templates containing AP sites (Gyenis et al, 2023; Hoeijmakers, 2009; Niedernhofer et al, 2018; Vermeij et al, 2014).

RAD51 may be especially important in mitigating oxidative DNA damage from metabolism and environmental exposure, which generates clustered, structurally complex lesions that strain canonical repair and impair DNA replication. These include free radical-induced tandem base modifications, chemically complex AP sites, and "dirty" strand breaks (Box et al, 1997; Yudkina and

Zharkov, 2025). By stabilizing stalled forks and protecting fragile abasic-containing intermediates, RAD51 can restrict unscheduled action of APE1 and other AP processing enzymes, steering repair toward accurate pathways. In this way, RAD51 likely cooperates with and helps coordinate BER at lesions that are otherwise poorly rectified reducing break formation when BER alone cannot efficiently resolve AP-site-containing structures (Basu et al, 2024; Paap et al, 2008).

HR-mediated protection of repetitive sequences, including telomeres and centromeres, might also matter for cell fitness in conditions where non-B DNA conformation and a high level of DNA methylation turnover might favor nuclease exposure of DNA (Salinas-Luypaert et al, 2025; Li et al, 2014). Human centromeres have been shown to accumulate HR proteins and other repair factors (Aze et al, 2016; Saayman et al, 2023; Scelfo et al, 2024a). Intriguingly, RAD51, PALB2 and BRCA2 are required to maintain centromere stability (Graham et al, 2025), also in non-replicating cells (Leung et al, 2024). It is therefore possible that HR factors protect against base damage and AP sites accumulation and processing within complex DNA sequences. Such function would be relevant not only for preventing cellular transformation, but also for preserving genome integrity in neuronal cells, which are known to accumulate chromosomes breakage from BER-generated AP sites occurring at gene enhancers (Wu et al, 2021).

## Conclusions

Although HR factors are traditionally recognized for their involvement in accurate DSB repair, emerging evidence reveals non-canonical roles for HR proteins in genome maintenance. In particular, RAD51 and its partners prevent genome damage accumulation and protect DNA from pathological degradation. These functions are mechanistically distinct from, and may predate, classical HR-mediated DSB repair. By binding distorted DNA structures, restraining nuclease activity, and cooperating with replication and repair factors, RAD51 mitigates diverse sources of stress, including those arising from epigenetic change. We are only beginning to define the full scope of HR protein functions. Systematically mapping how RAD51 and its partners act beyond DSB repair and intersect with non-HR pathways should reveal actionable vulnerabilities across disorders driven by dysfunctional DNA metabolism, from cancer to neurodegeneration and ageing.

## Peer review information

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

## Acknowledgements

We thank members of the Costanzo lab for critical discussions. The work leading to these results received funding from AIRC under IG 2023-ID.28725. AA is supported by the AIRC fellowship Italy Pre-Doc #31406.

## Author contributions

**Lorenzo Sassi**: Writing—original draft. **Andrea Martinez Marroquin**: Writing—original draft. **Salli Waked**: Writing—original draft. **Alessandra Ardizzoia**: Writing—original draft. **Vincenzo Costanzo**: Conceptualization; Writing—original draft; Writing—review and editing.

## Disclosure and competing interests statement

The authors declare no competing interests.

