## [Peer Review File · The EMBO Journal]

The Expanding Roles of Homologous Recombination Proteins in Genome Stability

Lorenzo Sassi, Andrea Martinez Marroquin, Salli Waked, Alessandra Ardizzoia, and Vincenzo Costanzo

Corresponding author(s): Vincenzo Costanzo (vincenzo.costanzo@ifom.eu)

Review Timeline:

Submission Date:	21st Jun 25
Editorial Decision:	21st Jul 25
Revision Received:	7th Oct 25
Accepted:	5th Dec 25

Editor: Hartmut Vodermaier

Transaction Report:

Prof. Vincenzo Costanzo
IFOM
DNA metabolism laboratory
Via Adamello 16
Milan 20139
Italy

21st Jul 2025

Re: EMBOJ-2025-121684
The Expanding Roles of Homologous Recombination Proteins in Genome Stability

Dear Vincenzo,

Thank you again for submitting your review article on genome protection roles of HR proteins. Two expert referees have now assessed the piece, and I am happy to say returned overall positive and appreciative comments (see below). Both of them raise several well-taken suggestions for specific improvements: expanding a bit deeper on certain aspects, improving the clarity of others, and thoroughly proofreading the study. Of note, both referees indicate that the use of figures needs to be improved, with the only two current figures remaining rather generic and not strongly synergizing with the text in demonstrating key new concepts. Please note that we would have the possibility of having draft figures re-drawn and touched up by graphic support from the journal side, but an original basic concept would be important.

In addition, a few editorial/formatting points would need to be taken care of during revision of the work:

- On the abstract page of the manuscript, please include 4-5 general keyword terms to enhance searchability.
- Please make sure that all relevant funding sources mentioned in the text are also entered into our submission system.
- Please include a Disclosure and competing interests statement, which we need also for review articles - for details, see <https://www.embopress.org/competing-interests>
- As we are switching from a free-text author contribution statement towards a more formal statement based on Contributor Role Taxonomy (CRediT) terms, please remove the present Author Contribution section and instead specify each author's contribution(s) directly in the Author Information page of our submission system during upload of the final manuscript. See <https://casrai.org/credit/> for more information.
- Importantly, please carefully go through the reference list, making sure that all references are complete with year/volume/page numbers, and that there are no duplicated references.
- Please adjust the section order as follows: Title page - Abstract - Keywords - Introduction - Acknowledgements (if any) - Disclosure and Competing Interests Statement - References - Figure Legends - Table(s) (if any)

Upon resubmission (accompanied by a brief point-by-point response and overview of changes made), I would myself go through the text one more time to copy-edit and smoothen out any last issues (such as unavoidable typos, unexplained abbreviations, passages unclear to non-experts...), and send the final figure drafts to our external graphics editors for redrawing and adaptation to journal style. Should you have any further questions regarding the revisions of text or graphics at this point, please don't hesitate to let me know.

Thank you once again for the opportunity to work with you on this, and I look forward to your revised manuscript!

With best regards,

Hartmut

*** PLEASE NOTE: All revised manuscripts are subject to initial checks for completeness and adherence to our formatting guidelines. Revisions may be returned to the authors and delayed in their editorial re-evaluation if they fail to comply to the following requirements (see also our Guide to Authors for further information):

1) Every manuscript requires a Data Availability section (even if only stating that no deposited datasets are included). Primary datasets or computer code produced in the current study have to be deposited in appropriate public repositories prior to resubmission, and reviewer access details provided in case that public access is not yet allowed. Further information:

embopress.org/page/journal/14602075/authorguide#dataavailability

9) To facilitate reproducibility and cross-laboratory adoption of methodologies, please structure the Materials & Methods section as outlined in our guide to authors, including a completed Reagents and Tools Table that can be downloaded from our author guidelines as well (<https://www.embopress.org/page/journal/14602075/authorguide#structuredmethods>).

10) Digital image enhancement is acceptable practice, as long as it accurately represents the original data and conforms to community standards. If a figure has been subjected to significant electronic manipulation, this must be clearly noted in the figure legend and/or the 'Materials and Methods' section. The editors reserve the right to request original versions of figures and the original images that were used to assemble the figure. Finally, we generally encourage uploading of numerical as well as gel/blot image source data; for details see: embopress.org/page/journal/14602075/authorguide#sourcedata

Further information is available in our Guide For Authors:

In the interest of ensuring the conceptual advance provided by the work, we recommend submitting a revision within 3 months (19th Oct 2025). Please discuss the revision progress ahead of this time with the editor if you require more time to complete the revisions. Use the link below to submit your revision:

Link Not Available

Referee #1:

General summary:

This review summarizes emerging studies on non-homologous recombination (HR) repair functions of BRCA1/2 and Rad51. It is

a well-written review with thoughtful discussion on how conventional HR proteins can have critical roles in replication fork protection, gap suppression and protection of abasic sites during nascent DNA synthesis. This review will be a valuable resource to the field. A few concerns are noted below:

Major concerns:

1. Figures: While the review is elegantly written, the figures are quite superficial. For instance, in Figure 1: Panel A: The schematic is unclear and can focus better on how Rad51 protects from fork uncoupling.
Panel B: Beyond inhibiting gap suppression, the panel can illustrate roles for Rad51 in preventing gap resection.
Panel C: Needs to be corrected- SMUG1 binds to abasic sites in dsDNA and not ss-DNA
PMID: 17537817
Panel D: The way the figure is drawn, it suggests that Rad51 converts conversion of base damage into abasic sites. Any direct evidence for this is lacking.

Similarly figure 2 can be elaborated to discuss fork reversal, binding of Rad51 to both single- and double-strand DNA to protect reversed forks and can include detailed illustration on the role of Rad51 in protecting AP sites MRN/CtIP cleavage.

2. Functions of Rad51 in BRCA1/2-independent fork reversal should be discussed (PMID: 37104614) and included in the Figure schematics. Also, it would be worthwhile to consider the possibility that gap enrichment in Rad51-deficient cells can be owing to loss fork reversal.

3. Given the review discuss a role for cleavage of abasic sites in enhancing genome stability it is imperative to discuss how factors that enhance abasic sites repair eg. ALC1 (PMID: 39068174; PMID: 39096699; PMID: 40238882) can significantly expand the therapeutic window of PARP inhibitors in BRCA-mutant cancers.

4. Clarify the following statements:

- (i) Abstract: "by Mre11 and related nucleases" should be by Mre11 and related nucleases at the replication forks.
- (ii) Page 4: "ss gaps can lead to replication fork collapse." Given nascent gaps are behind forks it is better to write "ss gaps on template DNA can lead to replication fork collapse."
- (iii) "Consistent with a direct role in DNA replication deficiencies in HR proteins have been" should be "Consistent with a direct role for HR protein in DNA replication, deficiencies in BRCA2 have been.."
- (iv) Page 8: The sentence is incomplete: "When such gaps occur in close proximity on both strands, they can lead to the cleavage of both"
- (v) Pg10: Typo: timidine
- (vi) Pg.15: Pena-Gomez et al., 2025: In this article, Abasic sites have been induced not in nascent strand but the template strand hence the phrase "ss-gaps containing AP sites is incorrect."

Minor suggestions:

- (i) Given the review and figures primarily focused on Rad51, authors can consider the following title: The Expanding Roles of Rad51 in Genome Stability.
- (ii) It would be worthwhile to direct the readers on recent reviews on how abasic sites are tolerated (PMID: 32417669) and what are the endogenous sources of abasic sites (PMID: 39096699).

Referee #2:

Manuscript:

The Expanding Roles of Homologous Recombination Proteins in Genome Stability

Sassi et al. present a concise and timely review that highlights the expanding roles of HR factors in counteracting endogenous DNA lesions, with particular emphasis on the formation of ssDNA daughter strand gaps. Nascent ssDNA daughter strand gaps have recently gained prominence as key intermediates in genome instability, especially in BRCA1/2-deficient tumors and their therapeutic vulnerabilities.

Centered around some pioneering work from their own lab and integrating emerging evidence from the growing literature in this area, this review offers a fresh perspective by discussing RAD51 as a genome 'surveillance' protein that actively prevents and protects against DNA damage during physiological replisome progression.

I enjoyed reading the review and fully support its publication. However, I recommend deepening several statements throughout the manuscript by citing appropriate references and elaborating-or, where appropriate, hypothesizing-on the underlying

mechanisms, rather than presenting them as general facts. In particular, a more detailed Figure of RAD51 interaction with abasic sites and the mechanistic links between DNA methylation, oxidative damage, and AP site formation would significantly strengthen the manuscript.

Moreover, explicitly defining the mechanistic distinctions in which HR operates-whether in the context of double-strand break repair or ssDNA gap suppression-as well as distinguishing its protective roles in shielding DNA lesions versus mediating fork reversal, would improve conceptual clarity. Additionally, clarifying whether these functions occur in the presence of an intact or broken replisome would further enhance the understanding of how RAD51 and other HR factors contribute to genome protection in a non-canonical manner.

Specific Comments:

Introduction section-

- Authors begin to explore the expanding roles of HR factors in genome protection by framing them as "non-canonical" functions. To enhance clarity for the reader, it would be helpful to include a line or two explicitly defining what is meant by "non-canonical" in this context. Specifically, does this refer to DSB-independent roles, structural functions at stalled replication forks, or surveillance activities that are distinct from or overlapping with the classical HR pathway involving DNA end resection, strand invasion, and repair synthesis? Additionally, this section would benefit from highlighting (and referencing) key studies that have employed separation-of-function mutants to mechanistically dissect the canonical DNA repair functions of HR proteins from their replication-associated or genome-protective roles. Such examples would provide valuable context and support for the conceptual distinction the authors are drawing.

Section: ssDNA gaps and abasic sites-

- This section is very well written, beginning with the pioneering work by Hashimoto et al. (2010), which provided the first evidence linking RAD51 and HR factors to the prevention of ssDNA gap accumulation during DNA replication. And more recently, again from the Costanzo lab, Hanthi et al. demonstrating a direct role for RAD51 in binding and protecting abasic sites, further expanding our understanding of RAD51 as a multifunctional genome integrity factor. While the section cites many of the important recent studies that have shaped our understanding of nascent ssDNA daughter strand gaps, it would benefit from acknowledging additional key papers published around the same time (PMIDs: 33203852; 34385259; 31607544; 33621493; 17379580), which contributed critical insights into the temporal dynamics of DSG repair, their spatial separation from the active replisome, and the pathological consequences of their persistence in BRCA- or HR-deficient in naturally or induced conditions.

- As suggested above, for this section, a more detailed figure illustrating the role of RAD51 in binding and protecting AP sites from nuclease activity by incorporating structural insights from Hanthi et al. would be highly valuable.

Section: Epigenetic origin of the abasic ssDNA gaps

- Statement: Although hypomethylation is typically linked to genomic instability through oncogene activation or loss of imprinting, it may also decrease a source of DNA lesions by restricting AP site formation. This reduction in damage may promote cell proliferation under specific conditions.

Please add references and, if possible, expand this in the context of RAD51/BRCA-defective cancers, as DNA hypomethylation at repetitive elements also promotes excessive homologous recombination and the formation of aberrant DNA structures.

Section: RAD51-mediated fork protection

- Although the authors state that BRCA2 modulates fork protection through mechanisms distinct from its role in homologous recombination-mediated repair, explicitly describing this distinction, ideally with reference to BRCA2's functional domains, would provide valuable clarity for readers.

- In a similar direction, expanding on whether and how RAD51's roles in fork reversal versus gap suppression are mechanistically separable would further strengthen the message.

- Additionally, a discussion of the status of active/stalled or broken replisome in the context of these distinct or overlapping genome-protective functions would be a valuable addition to better conceptualize the complexity of these mechanisms.

Other comments: A thorough proofreading is needed, as some statements currently end abruptly.

Referee #1:

General summary:

This review summarizes emerging studies on non-homologous recombination (HR) repair functions of BRCA1/2 and Rad51. It is a well-written review with thoughtful discussion on how conventional HR proteins can have critical roles in replication fork protection, gap suppression and protection of abasic sites during nascent DNA synthesis. This review will be a valuable resource to the field. A few concerns are noted below:

Our response: We thank this referee for the generous and positive assessment of our review. We're grateful that this referee found the discussion of BRCA1/2 and RAD51 beyond canonical HR to be clear and useful, particularly the sections on fork protection, gap suppression, and abasic-site safeguarding during nascent DNA synthesis. We appreciate the careful reading and the constructive concerns that follow. We have addressed each point in detail in our response and revised the manuscript to strengthen clarity, depth, and figure support accordingly.

Major concerns:

1. Figures: While the review is elegantly written, the figures are quite superficial. For instance, in Figure 1: Panel A: The schematic is unclear and can focus better on how Rad51 protects from fork uncoupling.

Panel B: Beyond inhibiting gap suppression, the panel can illustrate roles for Rad51 in preventing gap resection.

Panel C: Needs to be corrected- SMUG1 binds to abasic sites in dsDNA and not ss-DNA
PMID: 17537817

Panel D: The way the figure is drawn, it suggests that Rad51 converts conversion of base damage into abasic sites. Any direct evidence for this is lacking.

Similarly figure 2 can be elaborated to discuss fork reversal, binding of Rad51 to both single- and double-strand DNA to protect reversed forks and can include detailed illustration on the role of Rad51 in protecting AP sites MRN/CtIP cleavage.

Our response: Following the referee's suggestion, we have redrawn Figure 1. Panel A now depicts how RAD51 protects replication forks from uncoupling. Panel B illustrates how RAD51 prevents resection of replication-associated gaps.

Instead of Figure 1C, we now provide a new Figure 2. Panels A–C illustrate the RAD51 nucleofilament bound to abasic DNA, and Panel D presents original artwork showing SMUG1 accumulation on chromatin in the absence of BRCA2/RAD51. We have cited the referee's suggested reference on SMUG1 binding to AP sites in double-stranded DNA. We also highlight our recent evidence that SMUG1 accumulates on replicating DNA chromatin when BRCA2/RAD51 is absent and generates AP sites by excising uracil also on single-stranded DNA

(PMID: 39178838). Accordingly, Panel D reflects SMUG1's dual activity on dsDNA and ssDNA.

The previous Figure 1D has been removed. The new Figure 2D depicts the epigenetic pathway leading to AP-site formation in the absence of BRCA2/RAD51.

We have also replaced the former Figure 2 with a new Figure 3. As suggested, Panel 3A now shows how RAD51 binding to single-stranded DNA initiates fork reversal and how RAD51 engagement with the reversed double-stranded branch protects the remodeled fork from degradation. Panel 3B details the mechanism by which RAD51 protects AP sites from MRN-dependent cleavage. The molecular interaction of RAD51 with AP sites is shown in detail in Figure 2, Panels A–C.

2. Functions of Rad51 in BRCA1/2-independent fork reversal should be discussed (PMID: 37104614) and included in the Figure schematics. Also, it would be worthwhile to consider the possibility that gap enrichment in Rad51-deficient cells can be owing to loss fork reversal.

Our response: We now describe in detail, in Figure 3 and the accompanying text, the BRCA1/2-independent role of RAD51 and its paralogs in binding single-stranded DNA exposed at stalled forks to initiate fork reversal. We also discuss the possibility that impaired fork reversal contributes to the accumulation of ssDNA gaps.

3. Given the review discuss a role for cleavage of abasic sites in enhancing genome stability it is imperative to discuss how factors that enhancer abasic sites repair eg. ALC1 (PMID: 39068174; PMID: 39096699; PMID: 40238882) can significantly expand the therapeutic window of PARP inhibitors in BRCA-mutant cancers.

Our response: We thank the referee for highlighting the role of ALC1 (CHD1L) in AP-site repair. We have expanded the discussion to incorporate this point, added the suggested reference(s), and considered how ALC1-dependent chromatin remodeling at AP sites intersects with PARP1 activity and the therapeutic window of PARP1 inhibitors in BRCA-mutant cancers.

4. Clarify the following statements:

(i) Abstract: "by Mre11 and related nucleases" should be by Mre11 and related nucleases at the replication forks.

(ii)Page 4: "ss gaps can lead to replication fork collapse." Given nascent gaps are behind forks it is better to write "ss gaps on template DNA can lead to replication fork collapse."

(iii) "Consistent with a direct role in DNA replication deficiencies in HR proteins have been" should be "Consistent with a direct role for HR protein in DNA replication, deficiencies in BRCA2 have been.

(iv) Page 8: The sentence is incomplete: "When such gaps occur in close proximity on both strands, they can lead to the cleavage of both"

(v)Pg10: Typo: timidine

(vi) Pg.15: Pena-Gomez et al., 2025: In this article, Abasic sites have been induced not in nascent strand but the template strand hence the phrase" ss-gaps containing AP sites is incorrect."

Our response: We have now clarified all the indicated statements as requested.

Minor suggestions:

(i) Given the review and figures primarily focused on Rad51, authors can consider the following title: The Expanding Roles of Rad51 in Genome Stability.

Our response: We would like to emphasize that many of the mechanisms discussed in the review involve BRCA2. We have further expanded these sections to describe the relevant BRCA2 domains and separation-of-function mutants that affect HR-independent activities. The review also contains a substantial section on Fanconi anemia pathway factors in HR-dependent DSB repair. For these reasons, we have kept the original title, as it most accurately reflects the broad scope of the work.

(ii) It would be worthwhile to direct the readers on recent reviews on how abasic sites are tolerated (PMID: 32417669) and what are the endogenous sources of abasic sites (PMID: 39096699).

Our response: We have now mentioned and discussed the cited work.

Referee #2:

Manuscript:

The Expanding Roles of Homologous Recombination Proteins in Genome Stability

Sassi et al. present a concise and timely review that highlights the expanding roles of HR factors in counteracting endogenous DNA lesions, with particular emphasis on the formation of ssDNA daughter strand gaps. Nascent ssDNA daughter strand gaps have recently gained prominence as key intermediates in genome instability, especially in BRCA1/2-deficient tumors and their therapeutic vulnerabilities.

Centered around some pioneering work from their own lab and integrating emerging evidence from the growing literature in this area, this review offers a fresh perspective by discussing RAD51 as a genome 'surveillance' protein that actively prevents and protects against DNA damage during physiological replisome progression.

I enjoyed reading the review and fully support its publication. However, I recommend deepening several statements throughout the manuscript by citing appropriate references and elaborating-or,

where appropriate, hypothesizing-on the underlying mechanisms, rather than presenting them as general facts.

Our response: We thank the referee for the thoughtful and constructive assessment. We are pleased that the referee enjoyed reading our work. In response to the his/her recommendations, the manuscript has been revised to deepen the relevant statements, with claims now anchored to primary references and areas of uncertainty discussed as testable models. Sections on nascent ssDNA gap formation and fill-in now elaborate repriming, polymerase usage, and nucleolytic threats in BRCA1/2-deficient contexts. The RAD51 “surveillance” concept has been expanded by detailing filament dynamics, recruitment mechanism, and interactions with different domains of BRCA2 that coordinate gap suppression and the protection of abasic-site-containing intermediates. We have also updated the figures cross-referencing them to the relevant section of the text to guide interpretation. Finally, we have included more speculative arguments regarding poorly understood aspects of BRCA2 and RAD51 mediated control of genome stability independently of HR-mediated-DSB repair that warrant future studies as suggested. We trust these changes address the referee’s concerns and improve the precision and utility of the review.

In particular, a more detailed Figure of RAD51 interaction with abasic sites and the mechanistic links between DNA methylation, oxidative damage, and AP site formation would significantly strengthen the manuscript.

Our response: Prompted by the referee’s suggestion, we have added a new Figure 2 that details the mechanism of RAD51 binding and recruitment to DNA containing abasic sites. Panels 2A-C illustrate how Val273 occupies the void created by the abasic site within the damaged DNA duplex. We also include a new panel (2D) outlining a proposed pathway for AP-site formation from cytosine methylation. Finally, we have discussed the possibility that a subset of these modifications arises from oxidative stress.

Moreover, explicitly defining the mechanistic distinctions in which HR operates-whether in the context of double-strand break repair or ssDNA gap suppression-as well as distinguishing its protective roles in shielding DNA lesions versus mediating fork reversal, would improve conceptual clarity.

Our response: We have now better specified the different roles of HR proteins in gap suppression, protection and fork reversal in the text and by providing more detailed figures describing each of these aspects.

Additionally, clarifying whether these functions occur in the presence of an intact or broken replisome would further enhance the understanding of how RAD51 and other HR factors contribute to genome protection in a non-canonical manner.

Our response: We have clarified that the functions described pertain to an intact replisome. These activities act upstream of fork collapse to preserve fork integrity. We have better discussed that once collapse occurs, HR-dependent DSB repair contributes to repair the broken fork. We emphasize this distinction to explain why these mechanisms are essential under unchallenged conditions.

Specific Comments:

Introduction section-

- Authors begin to explore the expanding roles of HR factors in genome protection by framing them as "non-canonical" functions. To enhance clarity for the reader, it would be helpful to include a line or two explicitly defining what is meant by "non-canonical" in this context. Specifically, does this refer to DSB-independent roles, structural functions at stalled replication forks, or surveillance activities that are distinct from or overlapping with the classical HR pathway involving DNA end resection, strand invasion, and repair synthesis? Additionally, this section would benefit from highlighting (and referencing) key studies that have employed separation-of-function mutants to mechanistically dissect the canonical DNA repair functions of HR proteins from their replication-associated or genome-protective roles. Such examples would provide valuable context and support for the conceptual distinction the authors are drawing.

Our response: *We thank the referee for this helpful suggestion. The manuscript has been revised to define “non-canonical” functions explicitly as DSB-independent activities of HR proteins that act during physiological replication to prevent damage and preserve fork integrity, including surveillance at ssDNA gaps, structural stabilization at stalled forks, and protection of abasic-site-containing intermediates. We clarify how these activities are distinct from, yet partially overlapping with, classical HR steps such as end resection, strand invasion, and repair synthesis. In addition, we now highlight and reference key studies employing separation-of-function mutants in RAD51, BRCA1 and BRCA2 factors that mechanistically distinguish replication-associated and genome-protective roles from canonical DSB repair. We trust these additions address the request and improve clarity for readers.*

Section: ssDNA gaps and abasic sites

- This section is very well written, beginning with the pioneering work by Hashimoto et al. (2010), which provided the first evidence linking RAD51 and HR factors to the prevention of ssDNA gap accumulation during DNA replication. And more recently, again from the Costanzo lab, Hanthi et al. demonstrating a direct role for RAD51 in binding and protecting abasic sites, further expanding our understanding of RAD51 as a multifunctional genome integrity factor.

While the section cites many of the important recent studies that have shaped our understanding of nascent ssDNA daughter strand gaps, it would benefit from acknowledging additional key papers published around the same time (PMIDs: 33203852; 34385259; 31607544; 33621493; 17379580), which contributed critical insights into the temporal dynamics of DSG repair, their spatial separation from the active replisome, and the pathological consequences of their persistence in BRCA- or HR-deficient in naturally or induced conditions.

Our response. We thank the referee for pointing this out. The manuscript now cites the additional key papers (PMIDs: 33203852, 34385259, 31607544, 33621493, 17379580) at the relevant sections. We incorporated the findings described in these manuscripts into the text to emphasize the temporal dynamics of daughter-strand gap repair, the spatial separation of these gaps from the active replisome, and the pathological consequences of their persistence in BRCA/HR-deficient settings under both endogenous and induced conditions.

- As suggested above, for this section, a more detailed figure illustrating the role of RAD51 in binding and protecting AP sites from nuclease activity by incorporating structural insights from Hanthi et al. would be highly valuable.

Our response: We thank the referee for this valuable suggestion. We have added a new Figure 2 that illustrates the mechanism by which RAD51 binds and protects AP sites, incorporating the structural insights from our recent work (PMID: 39178838). The figure panels A-C depict the RAD51 nucleofilament engaged with abasic-site-containing DNA and shows how Val273 fills the void created by the abasic site, stabilizing the abasic lesion and helping shield the intermediate from nuclease attack. We believe this visual clarifies the protective mechanism and strengthens the section.

Section: Epigenetic origin of the abasic ssDNA gaps

- Statement: Although hypomethylation is typically linked to genomic instability through oncogene activation or loss of imprinting, it may also decrease a source of DNA lesions by restricting AP site formation. This reduction in damage may promote cell proliferation under specific conditions.

Please add references and, if possible, expand this in the context of RAD51/BRCA-defective cancers, as DNA hypomethylation at repetitive elements also promotes excessive homologous recombination and the formation of aberrant DNA structures.

Our response: We have revised the text to add the requested references and expanded the discussion on this topic in the context of RAD51/BRCA-defective cancers. Specifically, we now note that while global hypomethylation can drive genomic instability via oncogene activation and loss of imprinting, it may also reduce AP-site burden by limiting methyl-cytosine processing. We integrate evidence that hypomethylation at repetitive elements enhances hyper-recombination and favors the formation of aberrant DNA structures, which is particularly relevant in BRCA1/2-deficient settings. The revised section clarifies this duality and outlines the conditions under which reduced AP-site generation might transiently support proliferation.

Section: RAD51-mediated fork protection

- Although the authors state that BRCA2 modulates fork protection through mechanisms distinct from its role in homologous recombination-mediated repair, explicitly describing this distinction,

ideally with reference to BRCA2's functional domains, would provide valuable clarity for readers.

Our response: We have now detailed the individual domains of BRCA2 that support HR-mediated repair of DSBs and fork protection as suggested.

- In a similar direction, expanding on whether and how RAD51's roles in fork reversal versus gap suppression are mechanistically separable would further strengthen the message.

Our response: We have now expanded the discussion of RAD51's role in fork reversal vs gap suppression, mentioning their possible link and citing papers describing the available separation of function mutants the use of which could help further dissecting these aspects.

- Additionally, a discussion of the status of active/stalled or broken replisome in the context of these distinct or overlapping genome-protective functions would be a valuable addition to better conceptualize the complexity of these mechanisms.

Our response: We have now stated that these functions refer to active/stalled forks and not broken replisomes.

Other comments: A thorough proofreading is needed, as some statements currently end abruptly.

Our response: The paper has been carefully revised and these statements corrected.

Prof. Vincenzo Costanzo
IFOM
DNA metabolism laboratory
Via Adamello 16
Milan 20139
Italy

5th Dec 2025

Re: EMBOJ-2025-121684R
The Expanding Roles of Homologous Recombination Proteins in Genome Stability

Dear Vincenzo,

Thank you for submitting your final revised manuscript files. I am pleased to inform you that we have now accepted your review article for publication in The EMBO Journal!

With kind regards,

Hartmut
